# Evolutionary adaptation to juvenile malnutrition impacts adult metabolism and impairs adult fitness in *Drosophila*

Berra Erkosar[1†‡], Cindy Dupuis[1†], Fanny Cavigliasso[1], Loriane Savary[1], Laurent Kremmer[1§], Hector Gallart-Ayala[2], Julijana Ivanisevic[2], Tadeusz J Kawecki[1]*

[1]Department of Ecology and Evolution, University of Lausanne, Lausanne, Switzerland; [2]Metabolomics Unit, Faculty of Biology and Medicine, University of Lausanne, Lausanne, Switzerland

*For correspondence:
tadeusz.kawecki@unil.ch

[†]These authors contributed equally to this work

Present address: [‡]Foundation for Innovative New Diagnostics, Geneva, Switzerland; [§]Université Côte d'Azur – CNRS – INRAE, Institut Sophia Agrobiotech, Sophia Antipolis, France

Competing interest: The authors declare that no competing interests exist.

**Abstract** Juvenile undernutrition has lasting effects on adult metabolism of the affected individuals, but it is unclear how adult physiology is shaped over evolutionary time by natural selection driven by juvenile undernutrition. We combined RNAseq, targeted metabolomics, and genomics to study the consequences of evolution under juvenile undernutrition for metabolism of reproductively active adult females of *Drosophila melanogaster*. Compared to Control populations maintained on standard diet, Sselected populations maintained for over 230 generations on a nutrient-poor larval diet evolved major changes in adult gene expression and metabolite abundance, in particular affecting amino acid and purine metabolism. The evolved differences in adult gene expression and metabolite abundance between Selected and Control populations were positively correlated with the corresponding differences previously reported for sSelected versus Control larvae. This implies that genetic variants affect both stages similarly. Even when well fed, the metabolic profile of Selected flies resembled that of flies subject to starvation. Finally, Selected flies had lower reproductive output than Controls even when both were raised under the conditions under which the Selected populations evolved. These results imply that evolutionary adaptation to juvenile undernutrition has large pleiotropic consequences for adult metabolism, and that they are costly rather than adaptive for adult fitness. Thus, juvenile and adult metabolism do not appear to evolve independently from each other even in a holometabolous species where the two life stages are separated by a complete metamorphosis.

## Editor's evaluation

This important study pushes forward the understanding of a major question in evolutionary biology and human health: does juvenile malnutrition affect the performance of the adult? Combining experimental evolution in *Drosophila* with transcriptomics, metabolomics, and genomics datasets, it provides solid and compelling evidence that adult adaptation is constrained by adaptation at the larval stage. The work will be of interest to a broad audience including evolutionary biologists as well as human health researchers.

## Introduction

Nutrient shortage is an important and widespread ecological stressor, and a driver of natural selection on animal physiology. Juveniles are often the first to suffer from undernutrition as they are usually competitively inferior to adults. Furthermore, in most species, juveniles have no option to arrest their

development and wait for better times; they are forced to continue development despite famine. Prolonged nutrient shortage at a juvenile stage usually results in stunted adult size, but also often has profound effects on adult physiology that do not seem mediated by body size. They include changes in epigenetic marks (*Lillycrop and Burdge, 2015*), gene expression (*May and Zwaan, 2017*; *Szostaczuk et al., 2020*), metabolome (*Agnoux et al., 2014*), triglyceride accumulation (*Lukaszewski et al., 2013*; *Klepsatel et al., 2020*), blood pressure (*de Brito Alves et al., 2014*), behavior (*Akitake et al., 2015*), and longevity (*Davis et al., 2016*). In humans, prenatal undernutrition is correlated with negative consequences for diverse aspects of metabolic function and health at old age (*Roseboom et al., 2006*; *de Rooij et al., 2010*).

These physiological responses to developmental nutritional conditions are a form of phenotypic plasticity; that is, a change of phenotype induced by differences in the environment, with no change in genome sequence (*Scheiner, 1993*; *Bateson et al., 2004*). We know much less about whether and how adult physiology and metabolism evolve genetically over generations in response to natural selection driven by repeated exposure of the population to juvenile undernutrition. Some authors postulate that phenotypic plasticity in response to novel environmental conditions is usually adaptive, and that evolutionary change will mostly reinforce these initially plastic phenotypes ('genetic assimilation') (*Baldwin, 1896*; *Pigliucci and Murren, 2003*). Thus, plasticity would predict evolution. However, this prediction does not appear borne out by empirical data. In particular, experimental studies that compared plastic and evolutionary responses of gene expression to novel environments usually found little overlap between genes involved in the two responses; for those that did overlap, evolutionary change tended to counter rather than be in line with plasticity (*Yampolsky et al., 2012*; *Ghalambor et al., 2015*; *Huang et al., 2016*) for an exception, see *Josephs et al., 2021*. Thus, to understand how adult physiology is molded by exposure to juvenile undernutrition over evolutionary time, inferences based on phenotypically plastic responses should be complemented by the study of genetically based adaptations.

The particular vulnerability of juveniles to famine also raises the question of the degree to which metabolism and physiology of different life stages can evolve independently if they face different nutritional conditions. Anurans and holometabolous insects demonstrate that in the long run evolution can generate spectacular differences in morphology and physiology between juvenile and adult stages. This has led some authors to argue that metamorphosis of holometabolous insects allows the larval and adult phenotypes to be evolutionarily decoupled from each other (*Moran, 1994*; *Rolff et al., 2019*). This postulate requires a sufficient supply of genetic variants that affect relevant phenotypes in a stage-specific manner. However, surprisingly little is known to what degree this is the case, whether in holometabolous insects or other animals (*Collet and Fellous, 2019*). For example, while some enzymes in *Drosophila* have distinct larval and adult forms encoded by different genes, most metabolic genes are expressed at both stages (https://flybase.org/). Thus, independent evolution of larval and adult metabolism would require independent regulation of expression of the same genes. It may be more parsimonious to expect that most regulatory genetic variants would tend to affect gene expression across lifetime, constraining independent evolution of the juvenile and adult stages. If so, response to selection driven by juvenile undernutrition would lead to similar physiological change in larvae and adults, and these changes might be disadvantageous for adult performance, in particular under conditions of plentiful food.

Experimental data to address the above questions are scarce. Experimental evolution studies on *Drosophila* demonstrated that adaptation to poor larval nutrition (nutrient-poor diet or larval crowding) may be associated with changes in adult traits such as fecundity (*May et al., 2019*), lifespan (*Shenoi et al., 2016*; *May et al., 2019*), starvation resistance (*Kawecki et al., 2021*), pathogen resistance (*Vijendravarma et al., 2015*), or heat tolerance (*Kapila et al., 2021*). Changes in gene expression and metabolism underlying these correlated responses of adult phenotypes to selection for juvenile or larval undernutrition tolerance have not been explored. Insights into if and how selection driven by juvenile undernutrition shapes adult metabolism would shed light on the importance of evolutionary forces acting on development for understanding adult physiology, with potential implications for human health (*Neel, 1962*; *Prentice et al., 2005*).

Here, we use experimental evolution to study the consequences of evolutionary adaptation to larval undernutrition for gene expression, metabolite abundance, and reproductive performance of adult *Drosophila melanogaster* females. We compare six replicate populations (the 'Selected'

populations) evolved for over 230 generations on a very nutrient-poor larval diet, to six populations of the same origin maintained on a moderately nutrient-rich standard larval diet (the 'Control' populations). Importantly, in the course of experimental evolution, flies of both sets of populations were always transferred to fresh standard diet soon after emergence, maintained on this diet for 4–6 d, and additionally fed ad libitum live yeast before reproduction. Thus, every generation of Selected populations experienced a switch from the poor larval diet to the standard adult diet. Compared to the Controls, larvae of the Selected populations evolved higher viability, faster development and faster growth on the poor diet (*Kolss et al., 2009*; *Vijendravarma and Kawecki, 2013*; *Cavigliasso et al., 2020*). The Selected larvae appear to invest more in protein digestion (*Erkosar et al., 2017*), are more efficient in extracting amino acids from the poor diet (*Cavigliasso et al., 2020*), and are able to initiate metamorphosis at a smaller size (*Vijendravarma et al., 2012*), traits that likely contribute to their improved performance on the poor diet. The Selected populations also evolved reduced dependence on microbiota for development on poor diet (*Erkosar et al., 2017*), However, they show greater susceptibility to an intestinal pathogen (*Vijendravarma et al., 2015*) and are less resistant than Controls to starvation at the adult stage (*Kawecki et al., 2021*). Finally, the Selected and Control larvae also show divergence in metabolite abundance consistent with evolutionary changes in amino acid metabolism, with lower free amino acid concentrations and a higher rate of amino acid catabolism (*Cavigliasso et al., 2023*). Evolutionary divergence between Selected and Control populations is highly polygenic, involving over 100 genomic regions enriched in hormonal and metabolic genes (*Kawecki et al., 2021*). Thus, evolutionary adaptation to larval undernutrition in our experimental populations involved a complex suite of genomic and phenotypic changes.

Motivated by the general questions discussed above, in this study we address the following specific questions about the evolutionary divergence between the Selected and Control populations at the adult stage.

First, have the Selected and Control populations evolved major genetically based differences in adult gene expression and metabolite abundance, implying that evolutionary adaptation to larval undernutrition has significant consequences for the physiology of the adult stage? Which pathways have been affected?

Second, in view of the controversy about the relationship between plasticity and evolution summarized above, to what degree do these genetically based evolutionary changes in adult gene expression and metabolome resemble the phenotypically plastic responses to poor larval diet?

Third, are differences in adult gene expression and metabolite abundance between Selected and Control populations similar to those between Selected and Control larvae (reported elsewhere; *Erkosar et al., 2017*; *Cavigliasso et al., 2023*)? Such a similarity would be expected either if both life stages were exposed to similar selection pressures or if genetic variants favored by larval undernutrition had pleiotropic effects on gene expression at the two stages. As we argue in 'Discussion,' we find the latter interpretation more parsimonious, even it contradicts the notion that larval and adult gene expression patterns can be decoupled from each other in holometabolous insect (*Moran, 1994*; *Rolff et al., 2019*).

Fourth, do the metabolic profiles of Selected and Control flies differ along a similar axis as the profiles of fed versus starving flies? Selected flies are less resistant to starvation than Controls and the genomic architecture of divergence between them shares many genes with starvation resistance (*Kawecki et al., 2021*). This question addresses the link between these two findings, and the answer would aid in functional interpretation of metabolic changes evolved by the Selected populations.

Finally, do adult flies of the Selected populations perform better in terms of fitness than Controls when both are raised on the poor larval diet? This would be expected if most changes in their adult physiology evolved because they improve the performance of adults raised on poor diet. In contrast, if the metabolic changes observed at the adult stage primarily reflect pleiotropic effects of genetic variants that improve larval growth and survival on the poor diet, Selected adults should not have an advantage or even be inferior to Controls even when raised under the conditions under which the former evolved.

## Results

### Gene expression patterns point to evolutionary changes in adult metabolism

To explore plastic and evolutionary responses of adult gene expression to poor diet, we performed RNAseq on 4-day-old adult females from the Selected and Control populations raised on either larval diet. In this factorial design, the effect of larval diet treatment (poor versus standard) on the mean expression level is a measure of the plastic response. The difference between Selected and Control populations assessed in flies raised on the same diet is a measure of genetically-based evolutionary change (*Figure 1A*). Potential differences of the plastic responses between Selected and Control populations or the influence of diet on the expression of evolutionary divergence would be reflected in the evolutionary regime × diet interaction.

Prior to collection for RNAseq, the flies were transferred to standard diet upon emergence from the pupa and maintained in mixed-sex group; thus, they were mated and reproductively active. We focused on female abdominal fat body, the key metabolic organ combining the functions of mammalian liver and adipose tissue. It is in the fat body where metabolic reserves of glycogen and triglycerides are stored and mobilized, and where dietary nutrients are converted into the proteins and lipids subsequently transported to the ovaries for egg production (*Li et al., 2019*). Because a clean dissection of adult fat body is difficult, we performed RNAseq on the 'carcass,' consisting of the fat body together with the attached abdominal body wall and any embedded neurons, but excluding the digestive, reproductive, and most of the excretory organs.

Based on the estimation of true nulls (*Storey and Tibshirani, 2003*), the evolutionary regime (i.e., Selected versus Control) affected adult expression of about 26% of all 8701 genes included in the analysis while the larval diet affected about 45%. Allowing for 10% false discovery rate (FDR), we identified 219 genes as differentially expressed between Selected and Control populations and 827 between flies raised on standard or poor diet (*Supplementary file 1*). The magnitude of the changes in gene expression was moderate, with the significant genes showing on average about 1.76-fold difference (in either direction) between Selected and Control populations and about 1.30-fold difference between flies raised on poor versus standard diet. While the first three principal component axes appear driven by idiosyncratic variation among replicate populations and individual samples, PC4 and PC5 clearly differentiate the evolutionary regimes or larval diets (*Figure 1B*, *Figure 1—figure supplement 1*). The divergence of gene expression patterns between Selected and Control populations, as well as between flies raised on poor and standard diet, was supported by multivariate analysis of variance (MANOVA) on the first five principal components, jointly accounting for 67% of variance (Wilks' $\lambda$ = 0.052, $F_{6,5}$ = 15.1, $p$ = 0.0046 for regime and Wilks' $\lambda$ = 0.057, $F_{6,5}$ = 13.9, $p$ = 0.0055 for diet). We detected no interaction between the evolutionary regime and current diet either in the PCA-MANOVA (Wilks' $\lambda$ = 0.59, $F_{6,5}$ = 0.6, $p$ = 0.41) or in the gene-by-gene analysis (lowest $q$ = 99.9%). Thus, both the evolutionary adaptation and the phenotypically plastic response to larval diet experienced had substantial effects on adult gene expression patterns; however, these effects appeared largely additive (i.e., independent of each other).

The strongest statistical signal from Gene Ontology (GO) enrichment analysis on genes differentially expressed between Selected and Control flies points to genes involved in cuticle development and maturation, including several structural cuticle proteins (*Figure 1C*, *Supplementary file 2*, table A). Most of these genes are mainly expressed in epidermis (including in the tracheal system), and nearly all show higher expression in the Selected compared to Control populations. This may reflect differences in the relative amount of the epidermal tissues or in the rate of cuticle maturation. Several other enriched GO terms (phagocytosis, pigment metabolic process, iron ion homeostasis, response to fungus) include multiple genes expressed mainly in hemocytes; these genes are downregulated in Selected flies, suggesting lower abundance of hemocytes.

The other top enriched GO terms point to metabolic processes, notably oxidation/reduction, biosynthesis of purine compounds, and amino acid catabolism (*Figure 1C*). Differentially expressed genes involved in amino acid catabolism included enzymes involved in catabolism of arginine (*Oat*), serine (*Shmt*), branched-chain amino acids (*CG17896, CG6638, Dbct*), tryptophan (*Trh*), asparagine (*CG7860*), and tyrosine (*CG1461*). Thus, the signal of enrichment in amino acid metabolism was based on multiple differentially expressed genes distributed over multiple branches of amino acid metabolism. A few more genes involved in amino acid catabolism overexpressed in the Selected populations

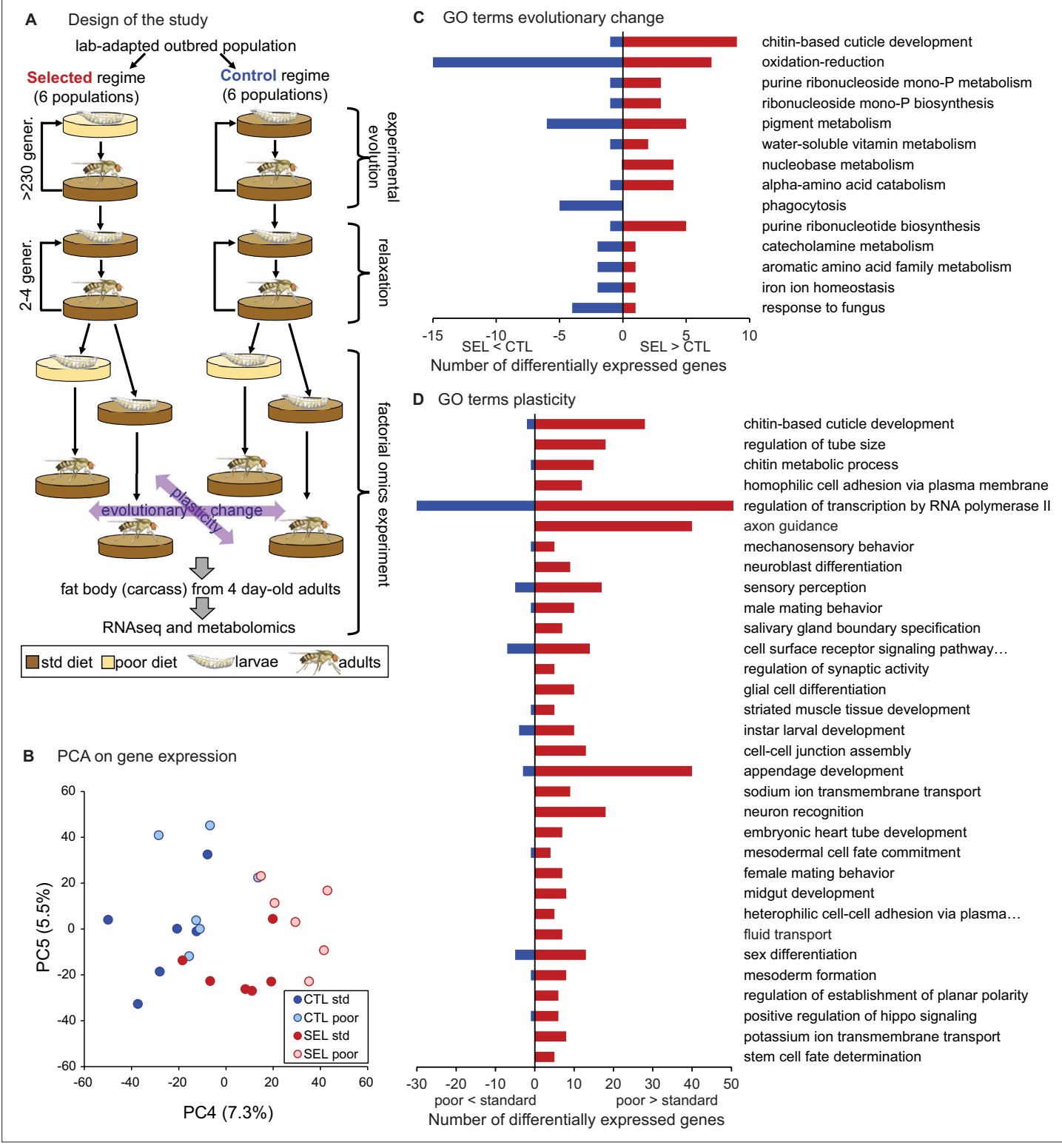

**Figure 1.** Phenotypic plasticity and genetically based evolutionary change in response to larval undernutrition both lead to divergence in gene expression patterns of adult *Drosophila* females. (**A**) Design of the experimental evolution and of the gene expression assay. The purpose of 'relaxation' was to eliminate the potential effects of (grand)parental environment. The purple arrows indicate the two main factors in the analysis and interpretation, the evolutionary history (evolutionary change), and the effect of larval diet on which the current generation was raised (plasticity). (**B**) Sample score plot of the fourth and fifth principal components on the expression levels of 8701 genes of flies from Selected (SEL) and Control (CTL) populations raised on poor and standard (std) larval diet. Each point represents a replicate population × diet combination (*Figure 1—source data 1*). For complete plots of

*Figure 1 continued on next page*

*Figure 1 continued*

PC1-6, see *Figure 1—figure supplement 1*. (**C**) Biological process Gene Ontology (GO) terms enriched (at $p < 0.01$) in genes differentially expressed between Selected and Control populations, with the number of genes upregulated (red) and downregulated (blue) in Selected populations. (**D**) GO terms enriched at $p < 0.01$ among genes showing a plastic response to diet. For details of the GO term enrichment results, see *Supplementary file 2*.

The online version of this article includes the following source data and figure supplement(s) for figure 1:

**Source data 1.** Principal component analysis of adult gene expression: the first six principal component scores of all populations on each diet.

**Figure supplement 1.** Plots of the first six principal components of the gene expression levels of adults from the Selected and Control populations raised on the standard and poor diet (*Figure 1—source data 1*).

link the aspartate–glutamate metabolism with purines synthesis pathway, which also showed a strong signal of enrichment in differentially expressed genes, and which we examine in more detail below.

## Metabolome analysis points to evolutionary changes in amino acid and purine metabolism

Differences in gene expression between Selected and Control flies suggested that evolutionary adaptation to larval diet affected adult metabolism. We therefore used a broad-scale targeted metabolomics approach to measure the abundance of key polar metabolites involved in multiple core pathways in central metabolism. As for gene expression, the analysis was carried out on the fat body with some adjacent body wall and neural tissue (the 'carcass') of 4-day-old females from Selected and Control populations, each raised on either poor or standard diet (additionally, we included a starvation treatment that is analyzed in a later section). After filtering for data quality, we retained 113 metabolites that were quantified (normalized to protein content) in all samples.

The first two principal components extracted from the metabolome data clearly differentiated the Selected versus Control populations, as well as flies raised on poor versus standard diet (*Figure 2A*; MANOVA on PC1 and PC2 scores, Wilks' $\lambda = 0.185$, $F_{2,9} = 19.8$, $p = 0.0005$ and Wilks' $\lambda = 0.055$, $F_{2,9} = 77.3$, $p < 0.0001$, for regime and diet, respectively). Of the 113 metabolites, 57 were found to be differentially abundant between Selected and Control flies at 10% FDR, and also 57 differentially abundant between flies raised on poor versus standard diet (*Supplementary file 3*). We found no interaction between these two factors either in the univariate analysis (no metabolite passing 10% FDR, lowest $q = 0.39$) or in the principal component analysis (PCA) (MANOVA, Wilks' $\lambda = 0.918$, $F_{2,9} = 0.4$, $p = 0.68$). Thus, both the evolutionary history of exposure to poor versus standard diet and the within-generation diet treatment had major effects on the metabolome, but, as for gene expression, these effects were largely additive. Therefore, we present the results as a heat map corresponding to the effects of these two factors on abundance of individual metabolites (the first two columns of heat maps in *Figure 2B*; the third column corresponds to the effect of starvation discussed in a later section).

Evolutionary adaptation to larval malnutrition resulted in a striking shift in proteinogenic amino acid concentrations in adult flies: eight essential amino acids were less abundant in the Selected than Control flies, while the reverse was the case for four non-essential ones (*Figure 2Bi*). Multiple purine compounds were less abundant in the Selected than Control flies, including both purine nucleobases and nucleosides, and the electron carriers NAD, NADP, and FAD; an exception is ADP, which was more abundant in the Selected flies (*Figure 2Bii*; ATP was not reliably quantified). No such general trend for reduced abundance was observed for pyrimidines (*Figure 2Biii*), suggesting that the pattern observed for purines is not mediated by changes in synthesis or degradation of nucleic acids.

Selected flies had lower levels of trehalose (the principal sugar circulating in hemolymph) and of lactate than Controls (*Figure 2Biv*), but higher concentrations of compounds linking glycerol with glycolysis, with no indication of differences in concentration of upstream and downstream glycolysis intermediates (*Figure 2Biv*). This suggests changes in fatty acid/triglyceride metabolism. While fatty acids were not included in the targeted metabolome analysis, we did quantify the abundance of acyl-carnitines, that is, fatty acid residues attached to carnitine, the molecule that transports them in and out of mitochondria. Compared to the Controls, the Selected flies show accumulation of medium chain (C6–C12) acyl-carnitines (*Figure 2Bv*). In contrast, acyl-carnitines with long-chain (C16–C18) residues, which are typical for insect triglycerides (*Stanley-Samuelson et al., 1988*), were less abundant in the Selected than Control flies (*Figure 2Bv*).

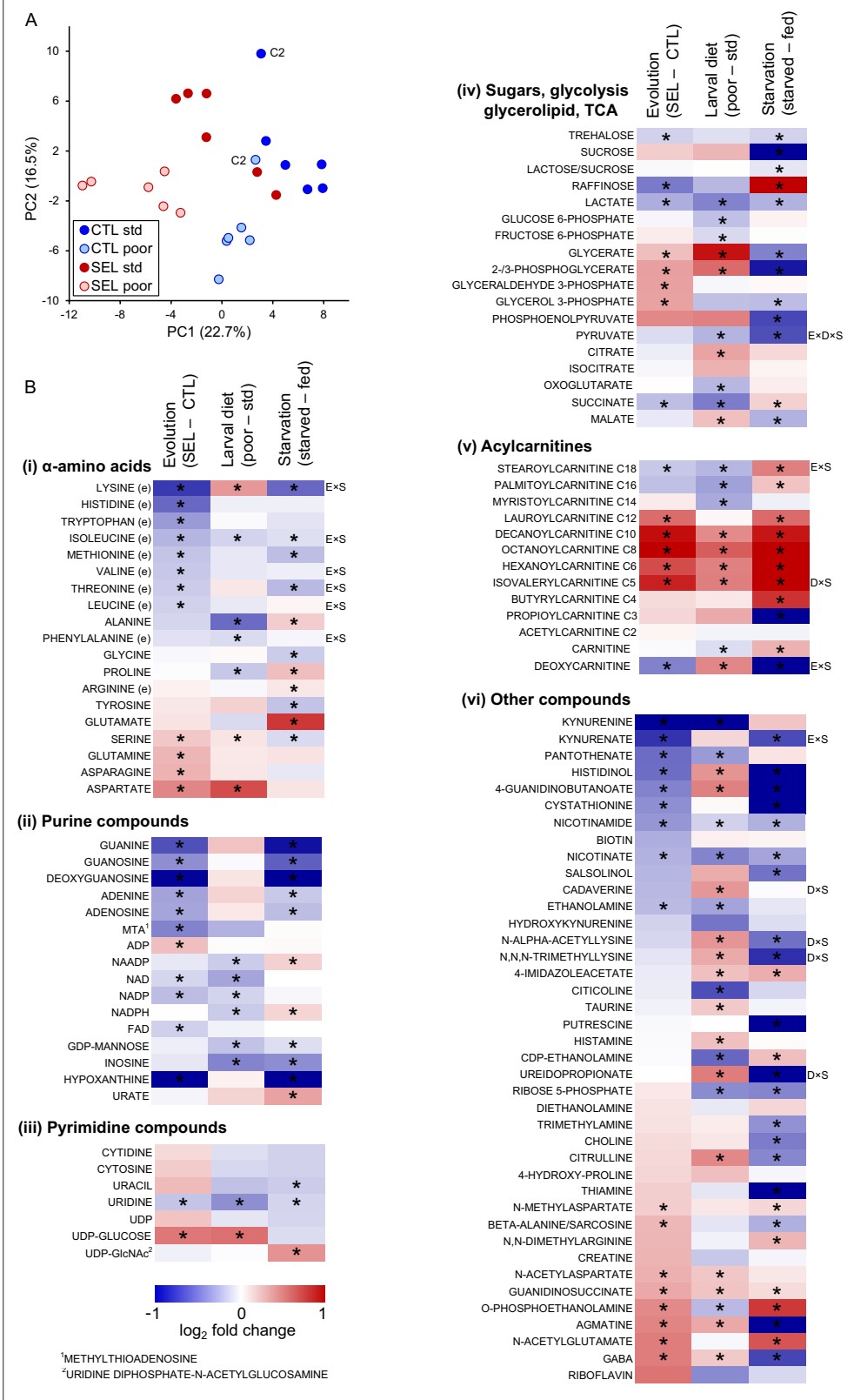

**Figure 2.** The effects of phenotypic plasticity and evolutionary change on adult metabolite abundance.
(**A**) Principal component score plot based on metabolic signatures composed of 113 robustly quantified
metabolites in flies from the six Selected and six Control populations raised on both poor and standard diet
(*Figure 2—source data 1*). The flies were in fed condition. Metabolite abundances for two replicate samples were

*Figure 2 continued on next page*

*Figure 2 continued*

averaged before the principal component analysis (PCA); thus, there is one point per population and diet. The 'C2' label indicates control population 2 on either diet; this population deviated from the other Control populations along the PC2 axis; however, we retained it for the univariate analysis. (**B**) The effects of experimental factors on abundance of individual metabolites. The first two columns of the heat maps indicate, respectively, the effect of evolutionary regime (least square mean difference Selected – Control) and the within-generation (phenotypically plastic) effect of larval diet (least square mean difference poor – standard diet) on the relative concentration of metabolites in fed flies. The third column indicates the main effect of the starvation treatment (least square mean difference between starved and fed flies) on metabolite concentrations in flies from both Selection regimes raised on both diets. Thus, red (blue) indicates that a compound is more (less) abundant in Selected, poor diet-raised and starved flies than in Control, standard diet-raised and fed flies, respectively. Asterisks indicate effects significant at $q < 0.1$. Annotations to the right of the heat maps indicate interactions significant at $q < 0.1$: E × S = evolutionary regime × starvation treatment, D × S = larval diet × starvation treatment, E × D × S = three-way interaction; (e) indicates an essential amino acid, C2–C18 for acylcarnitines refers to the length of the acyl chain. For estimates, effects, and statistical tests underlying the figure, see *Supplementary file 3*; original data are in *Supplementary file 8*.

The online version of this article includes the following source data for figure 2:

**Source data 1.** Principal component analysis of adult metabolite abundance in fed flies: the first four principal component scores of all populations.

---

A number of other metabolites showed differential abundance between Selected and Control flies (*Figure 2Bvi*). More than half of them are derivatives of amino acids, including intermediates in synthesis of NAD+ (kynurenine and kynurenate) and cysteine (cystathionine), neurotransmitters (GABA, N-methylaspartate [NMDA] and N-acetylaspartate [NAA]), and products of amino acid catabolism that do not appear to be functionally linked. Together with increased abundance of isovaleryl-carnitine (which carries a residue of deamination of branched-chain amino acids; *Figure 2Bv*), these results reinforce the notion that adaptation to the poor larval diet has been associated with wide-ranging changes in amino acid metabolism.

In an attempt to integrate metabolite, gene expression, and genome sequence changes associated with adaptation to the poor diet, we performed joint pathway analysis in Metaboanalyst v. 5.0 (*Xia et al., 2009*). In addition to the metabolites differentially abundant and genes differentially expressed between the Selected and Control populations, we included in this analysis 771 genes previously annotated to single-nucleotide polymorphisms (SNPs) differentiated in frequency between these two sets of populations (after 150 generations of experimental evolution; *Kawecki et al., 2021*). While most of the nine pathways identified in this analysis were only or mainly enriched in differentially abundant metabolites, two – purine metabolism and alanine, aspartate, and glutamate metabolism – involved a mix of metabolites and expression and SNP candidate genes (*Supplementary file 4*).

We therefore examined the evolutionary changes in these two pathways (imported from the KEGG database; *Kanehisa and Goto, 2000*). The alanine–aspartate–glutamate pathway contains multiple links among three of the four amino acids that are overabundant in the Selected flies. However, few of the many enzymes involved in those links showed a signal of differentiation between Selected and Control lines (*Figure 3A*). In contrast, purine metabolism showed a pattern of upregulation of multiple key enzymes involved in de novo purine synthesis in the Selected population, combined with overabundance of the three amino acids (glutamine, serine, and aspartate) that act as donors of four nitrogen atoms that form the basic purine structure (*Kastanos et al., 1997*; *Salway, 2018*) and provide (aspartate) a further amino group in the synthesis of AMP from IMP (*Figure 3B*). Despite this, compared to the Control flies, the Selected flies appeared depleted of nucleosides and nucleobases derived from IMP. We also found that multiple genes involved in AMP-ATP-cAMP conversion were associated with candidate SNPs differentiated in allele frequency between Selected and Control populations. While difficult to interpret functionally, this adds to the evidence that selection is driven by the dietary regime targeted purine metabolism.

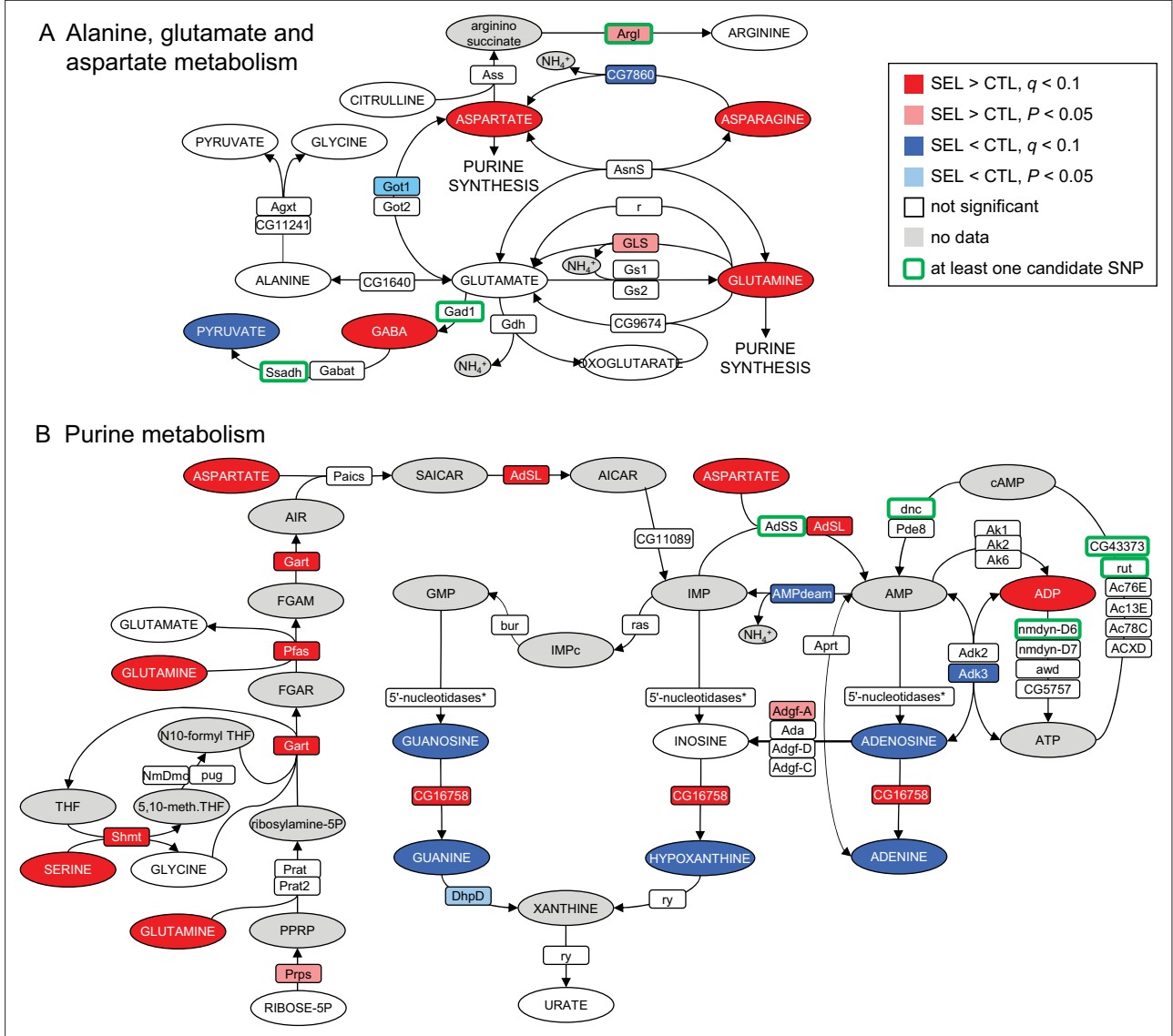

**Figure 3.** Imprint of evolutionary adaptation to poor larval diet on (**A**) alanine, glutamate, and aspartate metabolism and (**B**) purine metabolism pathways of adult female flies. Highlighted in color are metabolites differentially abundant and enzymes differentially expressed between the Selected and Control populations, as well as enzymes to which at least one single-nucleotide polymorphisms (SNP) differentiated in frequency between the sets of populations has been annotated. Enzymes differentially expressed at the nominal $p < 0.05$ but not passing the 10% false discovery rate (FDR) are also indicated. Metabolites in gray were either not targeted, not detected, or the data did not pass quality filtering. The pathways have been downloaded from KEGG (https://www.genome.jp/kegg/pathway.html). For the sake of clarity, some substrates and products of depicted reactions that were not differentially abundant or were absent from the data, as well as cofactors, have been omitted. Only genes expressed in female fat body have been included. *5'-nucleosidases: *cN-IIB*, *NT5E-2*, *Nt5B*, *veil*, *CG11883*; none was differentially expressed.

## Plastic response predicts in part evolutionary change of metabolome but not of gene expression

As was the case for Selected versus Control populations, the top GO term enriched for genes differentially expressed in flies raised on the poor versus standard larval diet treatment (corresponding to phenotypic plasticity) was 'chitin-based cuticle development' (*Figure 1D*; *Supplementary file 2, table B*). This is consistent with differences in body size: flies raised on the poor diet are much smaller than those raised on the standard diet (*Kolss et al., 2009*), and thus likely characterized by a greater surface-to-volume ratio and consequently a greater relative surface of cuticle on the carcass. This is, however, the only similarity between the effects of evolutionary regime and of the diet treatment for most enriched GO terms. No GO terms related to metabolism of amino acids, purines, or any other

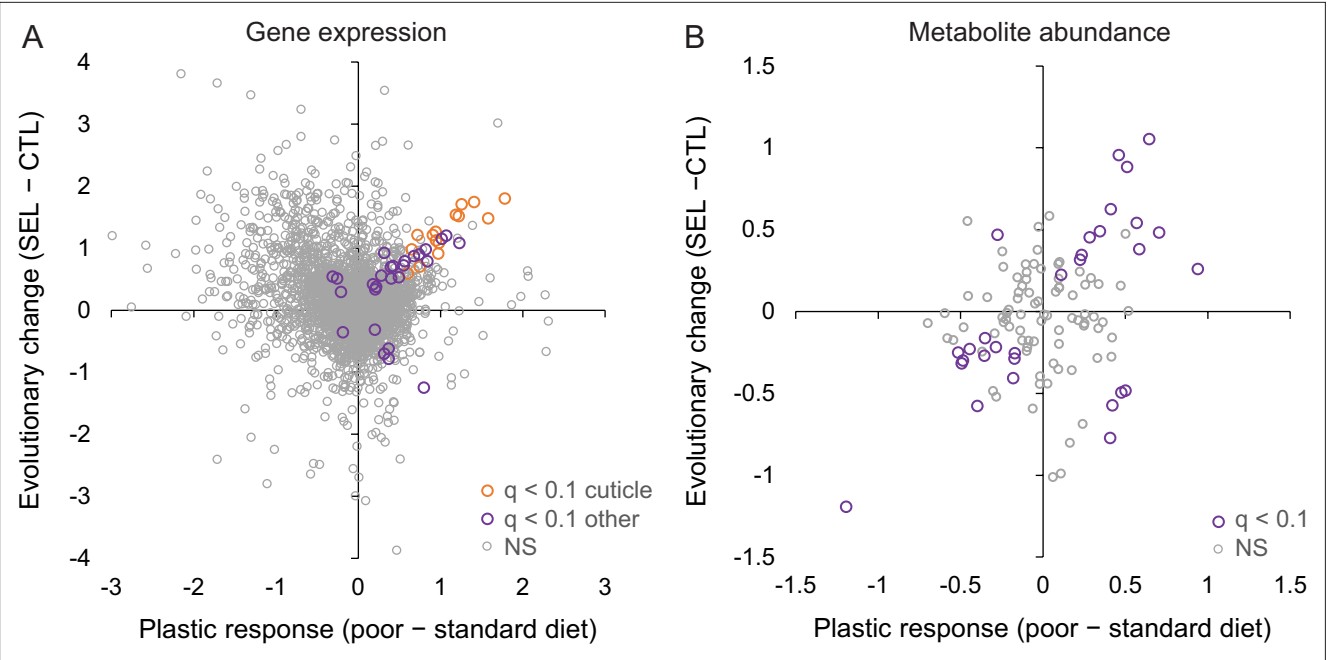

**Figure 4.** Relationship between phenotypic plasticity and genetically based evolutionary change of adult gene expression (**A**) and metabolite abundance (**B**) in response to larval diet. The plastic responses and evolutionary change for each gene and metabolite were estimated, respectively, as the main effect of larval diet and evolutionary regime in the factorial statistical models (see 'Materials and methods'). Color highlights genes and metabolites with significant effects ($q < 0.1$) for both diet and regime; in (**A**), orange symbols indicate significant genes involved in cuticle, integument, or tracheal development (*Figure 4—source data 1*).

The online version of this article includes the following source data for figure 4:

**Source data 1.** The estimated main effects of diet (the plastic response) and of evolutionary regime (the evolutionary change) on adult gene expression and metabolite abundance.

basic metabolic processes were significantly enriched for the effect of diet treatment. Instead, genes differentially affected by the diet treatment were enriched for many terms related to development, reproduction, nervous system, and behavior (*Figure 1D*; *Supplementary file 2, table B*).

This absence of parallelism between the evolutionary and phenotypically plastic responses to the poor larval diet is also visible at the level of individual genes. Only 42 genes were differentially expressed at 10% FDR for both of those factors; at least 15 of them are involved in integumentary (i.e., cuticle, epidermis, or tracheal) development, and most of the remaining ones play a role in the nervous system or transmembrane transport. Of 235 genes that passed the less stringent criterion of being differentially expressed at nominal (not adjusted) $p < 0.05$ for both factors, 123 showed the same sign of expression difference between poor and standard diet as between Selected and Control populations, and 112 showed the opposite sign, not different from what would be expected at random ($p = 0.13$, Fisher's exact test). Across all genes, the correlation between the main effect of diet and the main effect of evolutionary regimes was weakly negative ($r = −0.22$, $p < 0.0001$, $N = 8701$; *Figure 4A*), as was the corresponding correlation for genes annotated to metabolism (GO metabolic process, $r = −0.26$, $p < 0.0001$, $N = 4606$). Thus, except for genes involved in the cuticle maturation, the plastic (within-generation) response of adult gene expression pattern to larval undernutrition did not in general predict the evolutionary, genetically based response. If anything, evolution tended on average to oppose phenotypically plastic responses of gene expression to larval diet.

By contrast, the evolutionary change of metabolome driven by larval diet did to some degree parallel the phenotypically plastic response (correlation across all metabolites $r = 0.32$, $p = 0.0005$, $N = 113$; *Figure 4B*). Of the 30 metabolites that were significant at 10% FDR for both evolutionary regime and diet treatment, 25 showed the same sign of the difference between Selected and Control flies as the difference between flies raised on the poor versus standard diet (Fisher's exact test, $p = 0.0005$). In particular, compared to flies raised on standard diet, flies raised on the poor diet showed a lower abundance of long-chain acyl-carnitines but an excess of medium-chain ones (*Figure 2Bv*, middle

column), and a lower abundance of NAD+ and NADP (*Figure 2Bii*, middle column) and their precursors nicotinate and kynurenine (*Figure 2Bvi*, middle column), paralleling the patterns in Selected relative to Control flies (the left column of the respective heat maps). Some of the plastic effects of the poor diet treatment on sugar metabolism were also similar to those of evolutionary response, notably an underabundance of lactate and succinate, and an overabundance of UDP-glucose and of 2-/3-phosphoglycerate and glycerate, the latter accentuated by the underabundance of upstream and downstream glycolysis intermediates (*Figure 2Biv*).

More than a dozen modified amino acids and products of amino acid degradation were also affected, but these changes were not correlated with corresponding differences between the Selected and Control flies. Although several proteinogenic amino acids showed differential abundance depending on the diet treatment, these differences were also quite different from those due to the evolutionary regime and did not seem to show a consistent relationship with the properties of the amino acids. Finally, with the exception of inosine, the plastic response to diet did not detectably affect the abundance of purine nucleobases and nucleotides that do not contain a nicotinamide group (*Figure 2Bii*). These results are consistent with the absence of signal of an effect of the diet treatment on amino acid and purine metabolism in gene expression patterns presented above.

## Much of evolutionary change in gene expression and metabolome is conserved between life stages

The Selected and Control populations evolved under different larval diets but the adult diet they experienced was the same. Hence, while the divergence between them in larval gene expression and metabolite abundance is presumably driven by immediate nutrient availability, selection at the adult stage would likely result from the need to compensate for carry-over effects of larval diet. It does not seem likely that these two selection pressures would favor broadly similar changes in gene expression and metabolism. However, such a broad similarity would still be expected if the genetic

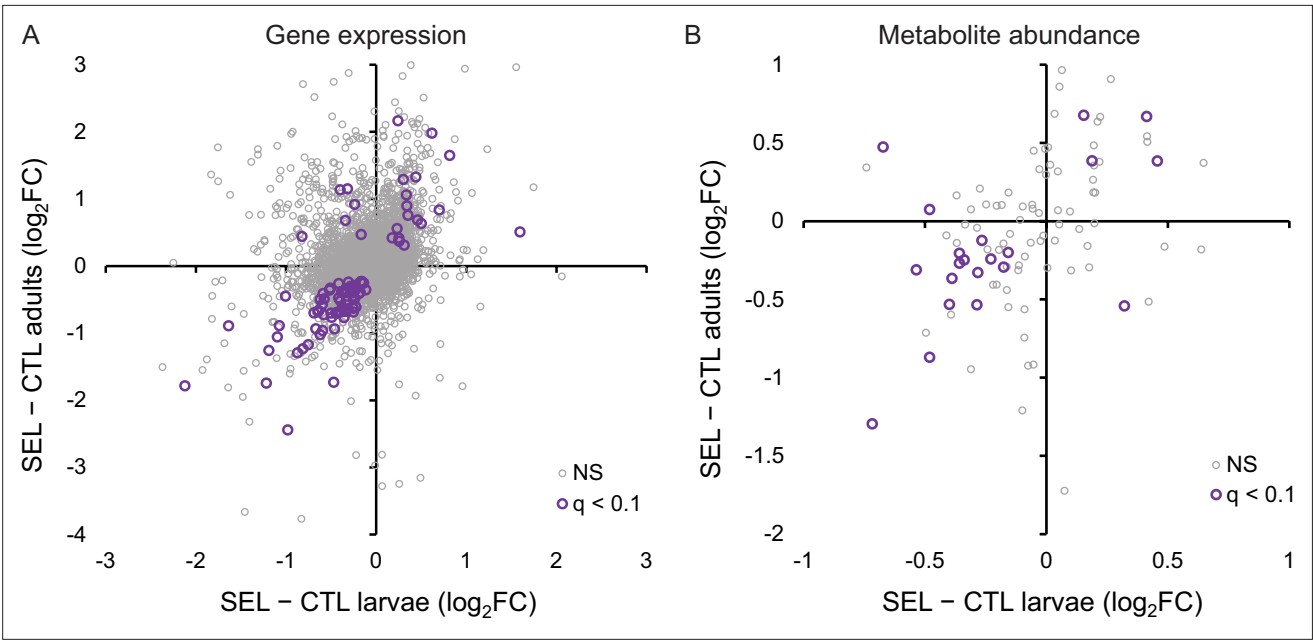

**Figure 5.** The relationship between the evolutionary change (i.e., the difference between the Selected and Control populations) in gene expression and metabolite abundance at the larval and adult stage. (**A**) Correlation of evolutionary changes in gene expression (all genes: $r = 0.35$, $N = 8437$, $p < 0.0001$; genes with $q < 0.1$: $r = 0.74$, $N = 84$, $p < 0.0001$). Five genes (not passing $q < 0.1$) are outside of the range of the plot. (**B**) Correlation of evolutionary changes in metabolite abundance (all metabolites: $r = 0.37$, $N = 97$, $p = 0.0002$; metabolites with $q < 0.1$: $r = 0.55$, $N = 21$, $p = 0.0091$). The estimates plotted are for larvae and adults raised on the poor larval diet (see 'Materials and methods'); $q < 0.1$ refers to genes or metabolites that pass the 10% false discovery rate (FDR) threshold for both stages; NS are the remaining genes or metabolites (*Figure 5—source data 1*, sheet A and B).

The online version of this article includes the following source data and figure supplement(s) for figure 5:

**Source data 1.** The relationship between evolutionary change in adults and the corresponding change in the larvae.

**Figure supplement 1.** The relationship between the phenotypically plastic responses of the metabolomes of larvae and adults to larval diet.

variants favored by selection on the larvae had similar effects across the stages. We therefore explored whether the divergence between Selected and Control populations in gene expression and metabolite abundance was correlated between the two life stages. To do so, we combined the present data from adult carcasses with previously published RNAseq and targeted metabolome data from whole third-instar larvae, collected at generation 190 (*Erkosar et al., 2017*) and generation 264 (*Cavigliasso et al., 2023*), respectively.

Of 424 genes that were differentially expressed at nominal $p < 0.05$ between Selected and Controls at both adults and larvae (*Supplementary file 5*), 377 (89%) showed the same direction of change on poor diet, more than expected at random (Fisher's exact test, $p < 0.0001$, *Figure 5A*). Eighty-four genes passed the more stringent criterion of 10% FDR at both stages; of those, 78 (93%) showed the same direction of change in the two stages (Fisher's exact test, $p < 0.0001$). The estimates of log-fold change were even correlated across all 8437 genes shared between the data sets ($r = 0.35$, $p < 0.0001$; *Figure 5A*). Thus, many of the differences between Selected and Control populations in the adult gene expression patterns paralleled those observed in the larvae.

Genes that were differentially expressed in Selected versus Control population in both adults and larvae were enriched in several terms related to transmembrane transport and iron homeostasis (*Supplementary file 2*, table D). Differences in gene expression of the larvae showed enrichment in lipid and carboxylic acid metabolism, as well as in a number of GO terms linked to cell proliferation and development, including that of the nervous system (*Supplementary file 2, table c*). The larval differentially expressed genes were not enriched in GO terms linked to amino acid or purine metabolism. However, 18 out of 99 genes in GO term 'alpha amino acid metabolic process' and 16 out of 107 in GO 'purine-containing compound biosynthetic process' were significantly different between Selected and Control larvae at 10% FDR (*Supplementary file 6*). Furthermore, the differences between Selected and Control populations across all genes in those GO terms were positively correlated between larvae and adults (amino acid metabolism: $r = 0.40$, $p < 0.0001$, $N = 94$; GO purine synthesis: $r = 0.47$, $p < 0.0001$, $N = 74$). Thus, while the amino acid metabolism and purine synthesis do not show disproportionate changes in gene expression in the larvae compared to other GO terms, the expression of multiple genes in those pathways has clearly been affected in a similar way as observed in adults.

Similarity between the effect of experimental evolution on larvae and adults was also apparent for the metabolome. Across all metabolites, the difference in metabolite abundance between the Selected and Control populations in the adults was positively correlated with the analogous difference in the larvae ($r = 0.37$, $p = 0.0002$; *Figure 5B*). This correlation was even more pronounced across the 21 metabolites that were differentially abundant at both stages ($r = 0.55$, $p = 0.009$); of those, 18 showed the same direction of change in larvae and adults, significantly more than expected by chance (Fisher's exact test, p = 0.011).

In contrast, in the same data set we detected no correlation between the direct (phenotypically plastic) responses of larval and adult metabolome to larval diet, whether across all metabolites ($r = 0.01$, $p = 0.89$) or across the 35 metabolites detected as differentially abundant in response to diet at both stages ($r = 0.17$, $p = 0.34$; *Figure 5—figure supplement 1*). Of that last group, 19 showed the same sign of change at both stages while 16 showed opposite signs, not different from random expectation ($p = 0.73$). This suggests that differences in metabolite abundance accrued at the larval can be erased within a few days of adult life. Hence, differences between Selected and Control adults in metabolite abundance are more likely to be mediated by differences in adult gene expression than by differential accumulation of metabolites during the larval stage.

## Metabolic profile of selected flies tends towards starved-like state

In addition to assessing the metabolome of flies directly sampled from the (standard) adult diet, we also assessed the metabolome of Selected and Control flies after they had been subject to 24 hr of total food deprivation (moisture was provided). The effect of the starvation treatment on the metabolic phenotype is also a phenotypically plastic response – to a different form of nutritional stress and one applied to adults rather than larvae. Our motivation was twofold. First, a comparison of fed and starved flies would reveal how the metabolome changes under an ongoing nutritional stress, and thus potentially help interpret metabolome changes driven by genetically based adaptation to recurrent nutrient shortage at the larval stage (i.e., the effect of evolutionary regime). Second, adult flies from

Selected populations are less resistant to starvation than Controls (*Kawecki et al., 2021*). Hence, we expected that the metabolome of the Selected and Control populations might throw some light on the mechanisms underlying these differences in starvation resistance.

As expected, 24 hr of starvation had a strong effect on fly metabolome. PCA cleanly separated starved from fed samples along the first PC axis (*Figure 6A*), while larval diet mainly defined the second PC axis. Of the 113 metabolites, 74 were detected to be differentially abundant (at 10% FDR) between fed and starved flies (in terms of the main effect of starvation treatment in the full factorial model, *Supplementary file 3*). Of 39 metabolites affected by both starvation treatment and the evolutionary regime, 29 showed the same direction of change between fed and starved flies as between Control and Selected flies, significantly more than expected by chance ($p = 0.006$, Fisher's exact test). In particular, in parallel to Selected flies relative to Controls, starved flies showed a decrease in multiple purine metabolites and two forms of vitamin B3 (nicotinate and nicotinamide), and an increase in medium-chain acyl-carnitines (*Figure 2B*). For some other metabolites, the effect of starvation differed markedly from that of the evolutionary regime. In particular, several intermediates of the glycolysis pathway were markedly less abundant in starved than fed flies (*Figure 2B*), whereas the surplus of acyl-carnitines also applied to those with long acyl chains. This possibly reflects a shift to catabolizing lipid stores for ATP generation following the depletion of carbohydrates.

The pattern for proteogenic amino acids also appeared rather different (*Figure 2B*). However, the focus on the main effects of starvation treatment on amino acid abundance is somewhat misleading because for 6 of the 10 essential amino acids the response to starvation differed between the Selected and Control flies. For all six, the sign of the interaction was the same: while the levels of all these amino acids decreased in Control flies when they starved ($q < 0.1$), their abundance declined less (lysine), remained similar (threonine, isoleucine), or even increased (phenylalanine, leucine, valine; $q < 0.1$) in starving Selected flies (*Figure 6B*). The other three metabolites with significant regime × starvation interaction showed different patterns (*Figure 6B*). There were also six metabolites with a significant larval diet × starvation or an interaction among all three experimental factors; they did not appear to be functionally connected or to share a common pattern (*Figure 6—figure supplement 1*).

These results imply that evolutionary adaptation of the Selected populations to larval undernutrition had consequences for the way their essential amino acid metabolism responds to food deprivation at the adult stage. Furthermore, even though in a normal fed state the Selected flies showed lower abundance of most essential amino acids, they could maintain or increase their levels during 24 hr of starvation. This suggests that their low concentrations in the fed state are not due to their absolute shortage but to differential allocation and use. In total, abundance of seven essential amino acids declined significantly upon starvation in Control flies, making the parallel with the difference between Selected and Control flies in fed state more evident. Indeed, the evolved differences in abundance of all proteogenic amino acids between Selected and Control flies were significantly positively correlated with the presumably ancestral response of standard diet-raised Control to starvation; the same was the case for purine compounds and acyl carnitines (*Figure 6C*).

Differential response of the Selected and Control populations to starvation is supported by regime × starvation interaction on PC1 scores ($F_{1,10} = 10.7$, $p = 0.0084$) – the metabolic state of Selected flies changed less upon starvation than that of Controls. This does not imply that the metabolome of Selected flies was more robust to starvation. Rather, fed flies from the Selected populations were situated closer to starved flies along the starvation-loaded PC1 axis than fed Control flies (*Figure 6A*; $F_{1,17.8} = 14.0$, $p = 0.0015$, GMM on PC1 scores); in the starved condition, they converged to a similar state ($F_{1,17.8} = 0.0$, $p = 0.99$; see *Supplementary file 7* for detailed statistics). Overall, the comparison of the (plastic) response of metabolome to starvation indicates that prominent aspects of the adult metabolic profile of Selected populations evolved in the direction resembling the effects of starvation.

## Adult cost of larval adaptation

The above results demonstrate that evolutionary adaptation to nutrient-poor larval diet has led to significant changes in adult metabolism. It is not clear, however, to what degree these changes may have evolved to improve fitness of adults that experienced poor diet as larvae, rather than being carry-over effects or trade-offs of changes driven by selection acting on the larval stage. If the former, adults from the Selected populations should perform better than those from the Control populations when both are raised under the conditions of the poor larval diet regime. If the latter, the fitness of

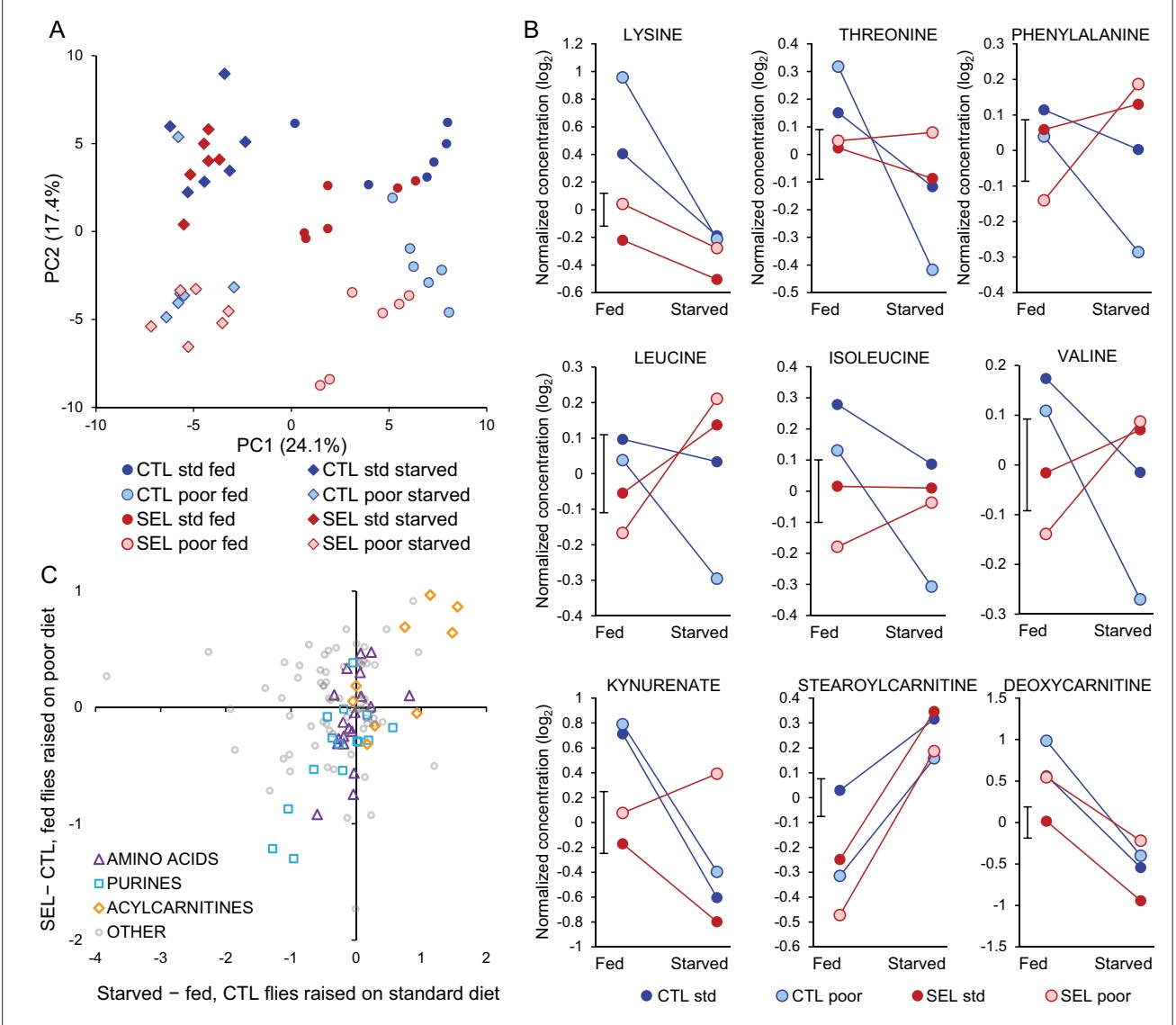

**Figure 6.** Effects of starvation on metabolite abundance. (**A**) Principal component score plot on metabolite abundance for all conditions (Selected versus Control, poor versus standard larval diet, fed versus starved flies). The two replicate samples from the same population × diet × starvation treatment combination were averaged (*Figure 6—source data 1*). (**B**) Metabolites that show significant ($q < 0.1$) interaction between the evolutionary regime and starvation treatment. Plotted are least-square means obtained from the full factorial general mixed model (*Figure 6—source data 2*). The bar next to the Y-axis indicates ± standard error of the least-square means (i.e., its width corresponds to 2 SEM); the standard error is identical for all treatments because the least-square means were estimated from the general mixed model and the design is balanced. For compounds showing other interactions, see *Figure 6—figure supplement 1*. (**C**) The relationship between the effect of 24 hr starvation (starved minus fed flies) on metabolite abundance in Control flies raised on the standard diet and the evolutionary change (Selected minus Control) quantified in fed flies raised on poor diet (*Figure 6—source data 3*). Three categories of metabolites of interest are highlighted; 'acylcarnitines' only include those with even-numbered acyl chain (resulting from catabolism of triglycerides) but not those with 3C and 5C chains (which are products of amino acid catabolism). Pearson's correlations: amino acids $r = 0.50$, $N = 19$, $p = 0.030$; purine compounds $r = 0.76$, $N = 16$, $p = 0.0006$; acylcarnitines $r = 0.74$, $N = 9$, $p = 0.022$; other metabolites $r = 0.02$.

The online version of this article includes the following source data and figure supplement(s) for figure 6:

**Source data 1.** Principal component analysis of metabolite abundance in fed and starved flies: the first four principal component scores of all populations under all conditions.

**Source data 2.** Abundance patterns of metabolites with interactions.

**Source data 3.** Relationship between response to starvation and the evolutionary change.

**Figure supplement 1.** Metabolites showing a significant three-way interaction evolutionary regime × larval diet × starvation (pyruvate) or the two-way interaction between larval diet and starvation treatment (the remaining five compounds).

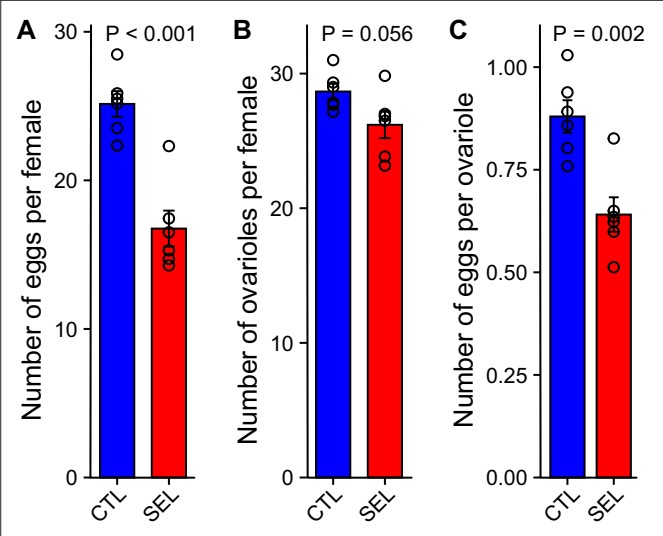

**Figure 7.** Selected populations pay a fecundity cost for larval adaptation to poor diet. (**A**) The number of eggs laid overnight by females from the Selected and Control (GMM; $F_{1,8}$ = 44.5, $p$ = 0.0002, N = 16 per evolutionary regime, 2–3 per replicate population). (**B**) The number of ovarioles (left + right ovary) per female (GMM; $F_{1,10}$ = 4.7, $p$ = 0.056, N = 72 per regime, six per replicate population). (**C**) The mean number of eggs per ovariole (one-way ANOVA; $F_{1,10}$ = 17.1, $p$ = 0.002, N = 6 populations per regime). The females experienced the dietary conditions, under which the Selected populations evolved: the larvae were raised on the poor diet; the adults were transferred to the standard diet and additionally provided live yeast 24 hr before oviposition. Bars are means, error bars correspond to ± SEM, the symbols to means of replicate populations (*Figure 7—source data 1*).

The online version of this article includes the following source data for figure 7:

**Source data 1.** Data on fecundity and ovariole number.

Selected adults raised on poor diet should be similar to or even lower than the fitness of Control adults developed on the same poor diet.

To test these alternative predictions, we compared the fecundity of females from Selected and Control populations raised on the poor larval diet, transferred to standard diet after eclosion, supplemented with ad libitum live yeast for 24 hr before oviposition, and allowed to oviposit overnight. These conditions mimicked those under which Selected populations have been maintained and propagated in the course of their experimental evolution; a female that laid more eggs in this time window would contribute proportionally more offspring to the next generation. The rate at which females can convert nutrients into eggs within this short time window is thus arguably a key measure of their fitness, and is inherently dependent on the metabolism of nutrients, in particular proteins, lipids, and nucleic acids. Despite having evolved under these conditions, Selected females laid only about 2/3 of the number of eggs produced by the Control females, whose ancestors did not face the poor larval diet (*Figure 7A*). Fly ovaries consist of a variable number of branches called ovarioles; in each ovariole, eggs are produced sequentially one by one. The rate of egg production may thus be limited by the number of ovarioles (*Schmidt et al., 2005*), which is determined during larval development (*Bergland et al., 2008*). Indeed, we found that poor diet-raised females of the Selected populations tended to have slightly fewer ovarioles than their Control counterparts (*Figure 7B*). However, even adjusted for the difference in ovariole number, the Selected flies produced substantially fewer eggs than Controls (*Figure 7C*).

*Drosophila* are 'income breeders' in that they directly convert acquired nutrients into eggs rather than relying on previously accumulated reserves (*O'Brien et al., 2008*). The fecundity difference thus implies that the Selected populations are less effective than Controls at converting dietary nutrients into the nutritional and/or generative components of the egg even when raised under the conditions under which the former but not the latter evolved during more than 200 generations.

## Discussion

### Costly correlated evolution of adult metabolism?

Experimental evolution under chronic larval undernutrition resulted in major shifts in gene expression patterns and metabolite abundance at the adult stage, replicable across six independent experimental (Selected) populations of *D. melanogaster*. This occurred even though adults of all populations experienced standard diet in the course of the experimental evolution. These evolved differences in adult gene expression and metabolite abundance between Selected and Control populations were broadly positively correlated with the corresponding differences between Selected and Control larvae. This was the case even though expression and metabolites were quantified in whole body in larvae but in dissected carcasses in adults, and despite these assays being performed dozens of fly generations apart. Therefore, the observed correlations between adult and larval evolutionary responses may have underestimated their actual similarity.

Even though adults of both Selected and Control populations experienced the same standard diet, they might still have been subject to differential selection on their adult metabolism. In particular, selection on Selected adults may have favored changes that compensate for lingering consequences nutritional hardship endured during the larval stage. However, it seems difficult to conceive why such compensatory responses of a well-fed adult should be mediated by similar physiological adjustments as those that promote larval growth under severe undernutrition. A more plausible explanation for the similarity in evolutionary changes between larvae and adults is that most genetic variants available for evolution affected gene expression of larvae and adults similarly. As a consequence, many of the adult gene expression changes would represent correlated responses to selection on the larval stage rather than the direct response to selection on adult performance (*Lande and Arnold, 1983*).

Such correlated responses may be detrimental to the performance of one or both life stages (*Collet and Fellous, 2019*). Consistent with this prediction, when raised on the poor diet and transferred to the standard diet as adults, the Selected flies were less effective than Controls in converting dietary nutrients into eggs, even though these were the conditions under which the former had evolved and to which the latter were exposed for the first time. (An analogous result on fecundity has been reported in another evolution experiment; *May et al., 2019*.) We have previously reported a lower fecundity of Selected females compared to Controls when both were raised on standard diet (*Kolss et al., 2009*); we interpreted that finding as a manifestation of reduced performance of Selected populations when raised on the standard larval diet. The present results imply that Selected adults perform less well than Control adults irrespective of the larval diet on which they have been raised. Other things being equal, a female's contribution to the next generation under the experimental evolution regime is proportional to the number of eggs she lays within a time window similar to that used to quantify fecundity. Thus, the fecundity reduction we found in Selected adults represents a significant fitness trade-off of the improved ability of the Selected larvae to survive and develop under extreme nutrient shortage.

### Amino acid catabolism and starved-like metabolic profile?

While Selected and Control adults differed in abundance of a variety of metabolites, the most striking pattern was the reduced abundance of 8 out of 10 essential amino acids in Selected flies. Fecundity in *Drosophila* is limited by essential amino acids (*Grandison et al., 2009*), in particular those in short supply relative to the needs of fly protein synthesis, which for a yeast-based diet are methionine and leucine (*Piper et al., 2017*). In addition, several essential amino acids (methionine, valine, isoleucine, and in particular leucine, all less abundant in Selected flies) act as signaling molecules that promote protein synthesis by activating TOR complex 1 (*Wolfson et al., 2016*; *Antikainen et al., 2017*; *Gu et al., 2022*). Most of the protein synthesis activity in adult female fat bodies is directed toward synthesizing egg proteins (*Piper et al., 2017*; *Gupta et al., 2022*). Thus, while we have not demonstrated a direct causal link, lower abundance of most essential amino acids in the Selected flies is consistent with their lower fecundity.

A reduction in free amino acid abundance could result from a lowered supply from nutrition and/or from a higher rate of their use for protein synthesis. However, these mechanisms should lead to a general depletion of free amino acids, which is not what we observed. Rather, in contrast to essential amino acids, several non-essential amino acids were overabundant in Selected flies; these overabundant amino acids are not enriched in dietary yeast relative to their need for *Drosophila* protein

synthesis (*Piper et al., 2017*). Thus, rather than being explained by differences in dietary amino acid acquisition or use in protein synthesis, the differences in amino acid abundance between Selected and Control populations appear consistent with differential use of amino acids in metabolism.

As adults, larvae of Selected populations also show lower abundances than Control larvae of multiple amino acids, including six essential, and a higher concentration of uric acid (*Cavigliasso et al., 2023*), an end product of purine metabolism and the main compound in which nitrogen is excreted in insects (*Salway, 2018*; *Cohen et al., 2020*). This suggests that Selected larvae catabolize amino acids and excrete nitrogenous waste products at a higher rate, a hypothesis further supported by their increased accumulation of the heavy isotope of nitrogen $^{15}$N (*Cavigliasso et al., 2023*). Although we found no difference in the levels of uric acid (urate) in the adult metabolome, it should be kept in mind that the adult metabolome was quantified carcasses, which only include fragments of Malpighian tubules, the excretory organs in which uric acid is synthesized (*Cohen et al., 2020*). It is notable that Selected flies show overabundance of the three amino acids – glutamine, aspartate, and serine – that contribute nitrogen atoms to purine/uric acid synthesis pathway, as well as overexpression of multiple genes involved in that process. Thus, while we do not have direct evidence for this, it remains possible that Selected flies catabolize amino acids and excrete uric acid at a higher rate than Controls also at the adult stage.

Increased catabolism of amino acids – sourced from autophagy – is one of the hallmarks of the physiological response to starvation (*Scott et al., 2004*). We found that several differences in metabolome between Selected and Control flies resemble the effects of starvation. In addition to reduction in abundance of several essential amino acids, this includes lower concentration of purine nucleotides and nucleosides, lower levels of trehalose and lactate, and higher abundance of acylcarnitines hinting at increased catabolism of fatty acids. Thus, the metabolic profile of Selected flies, in their normal fed and reproductively active state, resembles that of flies that have been starving. Consistent with the link with starvation, the genomic architecture of differentiation between the Selected and Control populations includes many candidate genes for starvation resistance (*Kawecki et al., 2021*). It is tempting to speculate that, as a by-product of genetic adaptation to poor larval diet, the Selected populations have become programmed to express a starved-like adult metabolic phenotype, which might explain their low fecundity.

## Relationship between phenotypic plasticity and evolutionary change

The evolutionary changes in gene expression did not in general recapitulate the phenotypically plastic responses of adult expression to larval diet. The abundance of a subset of metabolites did evolve in the direction that mimicked the plastic response, but this was not the case for most of the other metabolites. Congruence between the directions of phenotypically plastic response and evolutionary change driven by the same environmental factor has been interpreted as evidence for adaptive nature of the plastic response; conversely, where evolutionary change went in the opposite direction to the plastic response, the latter has been deemed maladaptive (*Yampolsky et al., 2012*; *Ghalambor et al., 2015*; *Huang et al., 2016*; *Josephs et al., 2021*). We believe this interpretation is not applicable to our results. First, as we argued elsewhere (*Cavigliasso et al., 2023*), a plastic response to a novel environment may be adaptive in terms of direction but overshoot the optimum phenotype; in such a case, an evolutionary change in the opposite direction would be favored despite the initial plastic response being adaptive. Second, the above interpretation assumes that evolutionary changes are mostly adaptive. However, as we have argued above, the evolutionary responses of adult gene expression and metabolism to larval undernutrition are likely to a large degree maladaptive costs of physiological adaptations favored at the larval stage. Thus, even plastic responses optimal from the viewpoint of adult fitness may have been reversed by the evolutionary change. These considerations imply that assessing the adaptive (or otherwise) nature of phenotypically plastic responses based on the direction of the evolutionary change may be misleading. Furthermore, our experimental results contradict the often-made assertion that phenotypic plasticity 'drives' evolutionary change (*Baldwin, 1896*; *Pigliucci and Murren, 2003*; *Moczek et al., 2011*; *Laland et al., 2015*).

## Evolutionary constraints on regulatory flexibility?

Like in all holometabolous insects, most larval tissues in *Drosophila* disintegrate during metamorphosis and the rest (e.g., brain) undergo extensive remodeling. Most of the adult structures and

organs are formed de novo from progenitor cells, resulting in an adult that is very different morphologically from the larvae, and physiologically specialized for a different function (reproduction rather than growth). In particular, cells of the larval fat body dissociate from one another and undergo autophagy and apoptosis during metamorphosis; the adult fat body develops anew from adult progenitor cells although possibly including some remaining larval fat body cells (*Li et al., 2019*). A major advantage of this complex development is thought to be that it decouples larval and adult gene expression, promoting independent evolution of larval and adult phenotypes (*Moran, 1994*; *Rolff et al., 2019*). Contrary to this notion, our results suggest that, at least on the scale of hundreds of generations, evolutionary changes in physiology driven by selection acting on larvae may have maladaptive pleiotropic effects on adult physiology. If this is the case in a holometabolous insect, such evolutionary non-independence of juvenile and adult physiology would likely be more pronounced in species with more developmental continuity between juvenile and adult tissues and organs.

Our study is thus relevant to understanding of constraints on the evolutionary refinement of physiology and life history of metazoans. Complex multicellularity crucially depends on the ability of the genome to express its genes differently in different cell types and life stages. The existence of specialized cells and organs that express greatly different yet highly coordinated and functional metabolic phenotypes from the same genome testifies to the power of regulatory evolution. On the other hand, there has been increased recognition that the ability of evolution to independently shape phenotypes of different life stages and sexes may be significantly constrained. Such constraints are expected to emerge from the complexity of gene regulatory and metabolic networks (*Wagner, 2011*; *Sorrells et al., 2015*; *Schaerli et al., 2018*). One manifestation of such constraints is 'intralocus sexual conflict' (sexually antagonistic pleiotropy), whereby simultaneous optimization of female and male phenotypes is hindered by constraints on independent evolution of gene expression in the two sexes (*Rice, 1984*; *Pischedda and Chippindale, 2006*; *Hollis et al., 2014*; *Veltsos et al., 2017*). Similarly, the developmental theory of aging postulates that gene expression and metabolism are optimized for maximizing performance at a young age and fail to adjust in later age in ways that could improve reproductive lifespan or healthspan (*de Magalhães, 2012*; *Gems and Partridge, 2013*), an idea increasingly supported by experimental data (*Carlsson et al., 2021*). This apparent metabolic inertia of aging individuals might be explained by selection at old age being weak (*Medawar, 1952*; *Hamilton, 1966*; *Partridge and Barton, 1993*). However, our results suggest similar evolutionary constraints linking the metabolism of juveniles and young adults in their reproductive prime, before the age-related decline in the strength of natural selection sets in *Hamilton, 1966*. Such constraints would hinder evolutionary optimization of juvenile and adult gene expression and metabolism if optima differ between the stages (*Collet and Fellous, 2019*).

## Materials and methods
### Diets, experimental evolution, and fly husbandry
Two diets were used for this study. The 'standard' diet consisted of 12.5 g dry brewer's yeast, 30 g sucrose, 60 g glucose, 50 g cornmeal, 0.5 g $CaCl_2$, 0.5 g MgSO4, 10 ml 10% Nipagin, 6 ml propionic acid, 20 ml ethanol, and 15 g of agar per liter of water. The 'poor' diet contained 1/4 of the concentrations of yeast, sugars, and cornmeal, but the same concentrations of the other ingredients. All experiments were carried out at 25°C and 12:12 hr LD cycle.

Six Selected and six Control populations were all derived from the same base population originally collected in 1999 in Basel (Switzerland) and maintained on the standard food for several years before the evolution experiment started in 2005 (*Kolss et al., 2009*). The six replicate populations per evolutionary regime constitute the main units of replication in this study; their number was limited by workload considerations. By the time of the experiments reported in this article, the Selected populations had been maintained on the poor larval diet for over 230 generations; the Control populations were maintained in parallel on the standard larval diet. Larval density was controlled at 200–250 eggs per bottle with 40 ml of food medium. Adults of both regimes were transferred to standard diet within 14 d of egg laying (sometimes a day or two later when not enough adults have emerged from the poor diet) and additionally fed live yeast 24 hr before egg collection to stimulate egg production. The target adult population size was 180–200 individuals.

Flies used in the experiments reported here were raised using similar procedures. Prior to all experiments reported in this study, all populations were raised on the standard diet for at least two generations to minimize the effects of maternal environment (*Vijendravarma et al., 2010*). For RNAseq and metabolome analyses, flies of both Selected and Control populations were raised on both larval diets. Eggs to establish the next generation were collected by allowing flies to oviposit overnight on orange juice agar sprinkled with live yeast. During the egg collection, the eggs were washed with water to remove any traces of diet at the surface – a procedure that also removes much of microbiota. To control the colonization by microbiota, eggs were re-inoculated using parental feces from a mix of adult flies from all 12 populations. These adults were left in a Petri dish and a wedge of food for 48 hr; they and the food were subsequently removed, the feces were washed from the surfaces of the Petri dish with PBS, filtered to remove eggs and debris, and adjusted to $OD_{600} = 0.5$. Each larval culture was established with 200 eggs transferred to a bottle with 40 ml of poor or standard diet, with 300 µl of the feces suspension pipetted on top; based on plating, this inoculum contains about $10^3$ CFU. All experiments were performed on females, aged 4–6 days old from eclosion. Selected larvae develop on the poor diet faster than Controls (*Kolss et al., 2009*; *Erkosar et al., 2017*). To ensure that the females from Selected and Control populations can be collected synchronously, we initiated the larval cultures in a staggered manner over several days, so that we could collect females from around peak of emergence and at the same time for Selected and Control populations despite the difference in developmental time. For RNAseq and metabolome quantification, adults of both sexes were collected around the peak of emergence, transferred to fresh standard diet, and allowed to mate freely for 3 d before females were collected. For fecundity and ovariole measurements, we collected virgin females.

## RNAseq on adult carcasses

Four-day-old mated female flies were collected in the morning and dissected in PBS. The abdomen was separated, and the gonads, the gut, and the bulk of Malpighian tubes were removed, leaving the 'carcass,' consisting of the abdominal fat body attached to the body wall, as well as any hemolymph, oenocytes, neurons, and fragments of Malpighian tubes that remained embedded in the fat body. Precise dissection of the adult fat body without disrupting it is very difficult, which is why carcass is typically used instead (e.g., modEncode project, *Brown et al., 2014*). From each of the 12 populations raised on either larval diet, we collected one sample of 10 carcasses, that is, 24 samples in total. The samples were snap-frozen in liquid nitrogen and stored at –80°C.

RNA was extracted from the carcass samples using 'Total RNA Purification Plus' by Norgen Biotek (#48300, 48400). cDNA libraries (TrueSeq Standard RNA) were generated and sequenced on two lanes of Illumina HiSeq4000 (single read, 150 bp) by the Genomic Technologies Facility of the University of Lausanne following manufacturer's protocols. Reads were mapped (pseudoaligned) to *D. melanogaster* reference genome BDGP6.79 using *kallisto* (*Bray et al., 2016*), with 22–31 million mapped reads per sample. Read counts per transcript feature output from *kallisto* were converted to counts per gene using *tximport* (*Soneson et al., 2015*). We filtered out genes with very low expression in that we only retained genes with read count per million greater than 2 in at least six samples. We further detected 45 pairs of genes with identical counts across all samples; from each pair we retained only the gene with a lower FBgn number, leaving 8701 unique genes.

Differential expression analysis was performed with *limma-voom* (*Law et al., 2014*; *Ritchie et al., 2015*), using *EdgeR* normalization (*Robinson et al., 2010*) implicit in the voom algorithm. Larval diet (poor versus standard), evolutionary regime (Selected versus Control) and their interaction were the fixed factors in the *limma* model; replicate population was a random factor modeled with *duplicateCorrelation* function of *limma*. Adjustment of p-values for multiple comparison was performed using Storey's FDR q-values (*Storey and Tibshirani, 2003*) as implemented in procedure MULTTEST option PFDR of SAS/STAT software v. 9. 4 (Copyright 2002–2012 by SAS Institute Inc, Cary, NC). Genes with expression different at $q < 0.1$ were considered significant for enrichment analyses. GO term enrichment analysis was carried out with bioconductor 3.12 package *topGO* v. 2.42.0 (*Alexa and Rahnenfuhrer, 2021*), using the *weight* method and Fisher's exact test. Because the tests for enrichment of different GO terms are highly non-independent, no meaningful method for calculating FDR exists (*Alexa and Rahnenfuhrer, 2021*); therefore, we report uncorrected p-values, focusing our interpretation on the top GO terms with $p < 0.01$.

Because the FDR threshold focuses on minimizing false positives, it leaves out many genes that have truly differed in expression. Thus, the number of genes that pass the FDR threshold greatly underestimates the number of genes that truly differ in expression, and the relationship between the two depends greatly on statistical power. Therefore, we also estimated the number of genes that differed in expression due to each factor in the analysis as the total number of genes minus the estimated number of 'true nulls' (*Storey and Tibshirani, 2003*).

To study to which degree the gene expression profiles of flies from the two regimes raised on the two larval diets were separable in a multivariate space, we performed PCA on the correlation matrix of the log-normalized expression data for all genes. To test for the separation of the samples in this multivariate space, we performed a MANOVA on the PCA scores, with regime, diet, and their interaction as the factors (using procedure GLM of SAS/STAT software).

## Larval gene expression analysis

The comparison of adult to larval gene expression differences was based on previously published data from an RNAseq study of whole larvae from the Selected and Control populations after about 190 generations of experimental evolution. All larvae were raised on the poor diet in that experiment in either germ-free state or colonized with a single microbiota strain at the high concentration of about $10^8$ CFU (*Erkosar et al., 2017*). Thus, the bacterial inoculation used for adult RNAseq (feces suspension containing about $10^3$ CFU) was intermediate between the two larval treatments. To maximize the compatibility of the adult and larval data sets, we remapped the larval reads using the same *kallisto* algorithm as for adult carcasses and analyzed the expression of the 11,475 genes with the same *limma-voom* approach, with evolutionary regime, microbiota treatment, and their interaction as the factors. We used the main effect of evolutionary regime from this study for the comparison with the adult results; using just data from the microbiota-colonized treatment led to qualitatively similar conclusions.

## Broad-scale targeted metabolomics

Metabolite abundance was measured using multiple pathway targeted analysis in the carcasses of 4-days-old mated females obtained as described above. From each population raised on each larval diet, we obtained two samples of 10 carcasses ('fed flies'). Two further samples of 10 carcasses per population and larval diet were obtained from females subject to 24 hr of starvation on nutrient-deprived agarose ('starved flies'), thus resulting in a three-way design (2 evolutionary regimes each with 6 populations × 2 current larval diets × fed versus starved flies, with 2 biological replicates, for a total of 96 samples, the number limited by cost). The number of samples was limited by the costs of metabolome analysis. The two samples for each population, larval diet and adult starvation treatment (i.e., fed or starved), were dissected by two experimenters, resulting in two experimental batches of 48 samples each. The two batches were also processed for metabolite extraction and analysis separately, on different days.

Extracted samples were analyzed by hydrophilic interaction liquid chromatography coupled to tandem mass spectrometry (HILIC-MS/MS) in both positive and negative ionization modes using a 6495 triple quadrupole system (QqQ) interfaced with 1290 UHPLC system (Agilent Technologies). Data were acquired using two complementary chromatographic separations in dynamic multiple reaction monitoring mode (dMRM) as previously described (*van der Velpen et al., 2019*; *Medina et al., 2020*). Data were processed using MassHunter Quantitative Analysis (for QqQ, version B.07.01/Build 7.1.524.0, Agilent Technologies). Signal intensity drift correction was performed on the pooled QC samples and metabolites with CV > 30% were discarded (*Dunn et al., 2011*; *Tsugawa et al., 2014*; *Broadhurst et al., 2018*). In addition, a series of diluted quality controls (dQC) were used to evaluate the linearity of metabolite response; peaks with correlation to dilution factor $R^2 < 0.75$ were discarded.

## Metabolome analysis

The peak area data for each compound were log-transformed, zero-centered, and Pareto-scaled separately for the two experimental batches. To test for differential abundance of single compounds, we fitted general mixed models (GMM) to these Pareto-scaled relative metabolite values using procedure MIXED of SAS/STAT software v. 9.4 (Copyright 2002–2012 by SAS Institute Inc). For each compound, we first fitted a full model, with evolutionary regime (Selected versus Control), larval diet (poor versus

standard), starvation treatment (starved versus fed), and all their interactions as fixed factors, and population nested within regime, diet × population and starvation × population as random factors. Inspecting the residuals from this model, we detected 13 data points (out of 10,848) with externally Studentized residuals greater in magnitude than ±4.0; these data points were removed as outliers and the model was refitted.

The residuals for 3 of the 113 compounds (creatine, hydroxykynurenine, and trehalose) deviated from normality at 10% FDR (Wilk–Shapiro test); we still report the results for these three compounds but they should be treated with caution. Because the main focus of the article is on the metabolism of flies in their normal fed state, we also analyzed the data from the 48 samples obtained from fed flies separately, fitting GMM with regime, diet, and their interaction, as well as experimental batch as fixed effects, and population nested in regime and diet × population interaction as random effects. The fixed factors in the GMM were tested with type 3 $F$-tests, using Satterthwaite method to estimate the denominator degrees of freedom. P-values for each factor were adjusted for multiple comparison as for differentially expressed genes. To illustrate the effects of the experimental factors (in particular, the interaction between the effects of evolutionary regime and starvation; *Figure 6B*), for some metabolites we plotted the estimated marginal means from the GMM. Because the design was balanced and the error variance was estimated from the model, all marginal means had the same standard error; to reduce the clutter in the plots, we plotted this common standard error as a single error bar rather than adding such bars to each symbol representing the mean.

To visualize and test for multivariate differentiation of the metabolome, we performed PCA on log-abundances of the 113 metabolites. The values from the two batches were averaged before this analysis so there was one point per population × diet combination × starvation treatment. As for gene expression patterns, sample scores from the PCA were tested for the effects of experimental factors and their interaction in a MANOVA implemented in procedure GLM of SAS/STAT software.

We also explored the relationship between the evolutionary change in metabolite abundance and the response to starvation for several categories of compounds. We specifically examined if the difference in metabolite abundance between Selected and Control flies raised on poor food, in their normal fed state, was correlated (Pearson's $r$) with the difference between starved and fed Control flies raised on the standard diet. These two variables are functions of non-overlapping sets of measurements, thus avoiding spurious correlations due to non-independence of errors.

## Fecundity and ovariole number

Fecundity was only assayed in females raised on poor larval diet, as described above. Male genotype and in particular the seminal fluid proteins males transfer to females during mating affect the short-term fecundity of the female. To ensure that any differences in egg number among populations are driven by female physiology and not by differences between males to which those females were mated, females from all populations were mated to males from a single laboratory population, originally collected from another site in Switzerland (Valais) in 2007. From each population, we collected 25–35 virgin females at the peak day of emergence and transferred them to standard food. Three days later, we split the females into replicates of 10–12 females (2–3 replicates per population depending on the total number of females remaining alive) and placed them together with 10–12 young males on orange juice-agar medium supplemented with live yeast; they were allowed to feed and mate for 24 hr. Subsequently, males were discarded (to prevent potential fecundity reduction due to male harassment) and the groups of females were allowed to oviposit overnight in new bottles with fresh orange juice-agar supplemented with live yeast. Eggs were washed from the medium surface with tap water, collected on a fine nylon mesh, and transferred to a well of a 12-well cell culture plate containing 3 ml of 1% sodium dodecyl sulfate (SDS, Sigma) to facilitate egg dispersion. A photograph of each well was taken under a Leica stereomicroscope with a Canon 60D camera with manual exposure programming, automatic white balance, 3.2 s of exposure time, and ISO 400.

The number of eggs was estimated automatically with Codicount ImageJ plugin, following *Perez, 2017*. This approach is based on automatically quantifying the area corresponding to the eggs on the image, based on color contrast between the eggs and the background. This total egg area was then converted to an estimate of the number of eggs by dividing it by the area of a single egg, estimated from a separate sample (the same standard egg area was used for all images). Thus, the estimated number of eggs was divided by the number of females in a particular replicate. In one replicate, 1 of

10 females was found dead at the end of oviposition; for the analysis, we set the number of females in this bottle at 9.5.

Flies were raised on the poor larval diet. While the number of ovarioles is fixed by the time of emergence, they are better visible and easier to count if filled with developing eggs (*Bergland et al., 2008*). We therefore allowed 25 freshly emerged females to mate with 25 males on standard diet for 3 d and subsequently to feed on ad libitum live yeast for 2 d, before collecting and storing them at –80°C until dissections. Six females from each population were haphazardly chosen for ovary dissection. We dissected the ovaries by separating the abdomen and cutting its posterior end, opening the abdomen laterally, and removing the ovaries. We then dipped the ovaries briefly in a solution of crystal violet to improve visual contrast, opened them, and counted the number of ovarioles in each ovary. Dissections and counting were done blindly with respect to the identity of the sample. To normalize the fecundity of each population relative to ovariole number, we divided the mean number of eggs per female in this population by the mean number of ovarioles.

The number of replicates was based on practical considerations, no formal power analysis has been conducted. The estimation of egg number by counting ovarioles was performed blindly with respect to the identity of the sample. Egg number and ovariole number per female were analyzed with a GMM, with regime as a fixed factor and population nested in regime as a random factor. For the number of eggs per ovariole, we only had one data point per population; these were compared between regimes with a one-way linear model.

## Acknowledgements

We thank D Promislow for ideas, T Teav at Metabolomics Platform at UNIL for his contribution to sample preparation and data processing, and three reviewers for their constructive comments on a previous version of the manuscript.

## Additional information

### Funding

| Funder | Grant reference number | Author |
| --- | --- | --- |
| Swiss National Science Foundation | 31003A_162732 | Tadeusz J Kawecki |
| Swiss National Science Foundation | 310030_184791 | Tadeusz J Kawecki |
| Research funds of the University of Lausanne | | Tadeusz J Kawecki |

The funders had no role in study design, data collection and interpretation, or the decision to submit the work for publication.

### Author contributions

Berra Erkosar, Conceptualization, Data curation, Formal analysis, Investigation, Methodology, Writing – original draft, Writing – review and editing; Cindy Dupuis, Investigation, Writing – review and editing; Fanny Cavigliasso, Formal analysis, Investigation, Writing – review and editing; Loriane Savary, Investigation; Laurent Kremmer, Investigation, Methodology; Hector Gallart-Ayala, Conceptualization, Formal analysis, Investigation, Methodology, Writing – review and editing; Julijana Ivanisevic, Conceptualization, Methodology, Writing – review and editing; Tadeusz J Kawecki, Conceptualization, Formal analysis, Funding acquisition, Visualization, Writing – original draft, Project administration, Writing – review and editing

### Author ORCIDs

Berra Erkosar http://orcid.org/0000-0003-1152-6772
Fanny Cavigliasso https://orcid.org/0000-0002-7764-4934
Hector Gallart-Ayala https://orcid.org/0000-0003-2333-0646
Julijana Ivanisevic http://orcid.org/0000-0001-8267-2705

Tadeusz J Kawecki ⬤ http://orcid.org/0000-0002-9244-1991

**Decision letter and Author response**
Decision letter https://doi.org/10.7554/eLife.92465.sa1
Author response https://doi.org/10.7554/eLife.92465.sa2

## Additional files

### Supplementary files

• Supplementary file 1. Results of univariate analysis of gene expression in adult female carcass performed in limma-voom.

• Supplementary file 2. Results of GO term enrichment for 'biological process' on genes expressed in female adult carcass. (A) Genes differentially expressed between Selected and Control flies. (B) Genes differentially expressed in flies raised on standard versus poor larval diet. (C) Genes differentially expressed between Selected and Control populations both in adults and larvae. (D) Genes differentially expressed between Selected and Control populations in adults but not in larvae.

• Supplementary file 3. Results of univariate analysis of metabolome with general mixed model. (A) Least-square means for all combinations of the three experimental factors: evolutionary regime (Control versus Selected), larval diet (standard versus poor), and adult starvation treatment (fed versus starved). The LS means and standard errors (SE) were obtained from the full GMM model. They are expressed on a zero-centered log2 scale. (B) Least-square means contrasts corresponding to *Figure 2B*: Regime(Fed) = Selected – Control in fed condition; Diet(Fed) = flies in fed condition raised on standard diet – flies in fed condition raised on poor diet; Starv = starved flies – fed flies across both regimes and diets. (C) Tests of significance (df, F, nominal P, q = adjusted P) from separate analysis on flies in fed condition only. (D) Tests of significance from the full model on both fed and starved flies. Denominator degrees of freedom (df) were estimated with the Satterthwaite method and thus vary across compounds depending on the magnitude of variance components corresponding to random factors.

• Supplementary file 4. Results of joint pathway analysis on metabolites significantly different between Selected and Control flies (in fed state).

• Supplementary file 5. Genes that show differential expression between Selected and Control populations (at raw p < 0.05) at both larval and adult stage.

• Supplementary file 6. Analysis of the effect of evolutionary regime on the expression of genes involved in amino acid metabolism and purine synthesis in the larvae.

• Supplementary file 7. MANOVA on the first two principal component scores from the principal component analysis on metabolite abundance in fed and starved flies.

• Supplementary file 8. Original data on metabolite abundance. The table includes the original estimates of peak area (in arbitrary units) normalized to the sample protein content, and log10-transformed, zero-centered, and Pareto-scaled values used for the analysis (zero-centering and Pareto-scaling was done separately for each batch). The table includes the 13 outlier data points that were removed from the final analysis because their Studentized residuals exceeded ±4; these outliers are identified in the last column.

• MDAR checklist

### Data availability

The raw and processed data from the RNAseq on adult carcasses are available from NCBI GEO (accession number GSE193105). Raw data for the previously published larval RNAseq are available from NCBI SRA (accession numbers SAMN07723150-SAMN07723173). Previously published larval metabolome data are available as supplementary material to *Cavigliasso et al., 2023*. The adult metabolite abundance data are provided in Supplementary file 8, fecundity and ovariole data in Figure 7-source data 1.

The following dataset was generated:

| Author(s) | Year | Dataset title | Dataset URL | Database and Identifier |
|---|---|---|---|---|
| Kawecki TJ, Erkosar B, Dupuis C, Savary L | 2022 | Evolutionary and phenotypically plastic response of adult gene expression to larval undernutrition in *Drosophila melanogaster* | https://www.ncbi.nlm.nih.gov/geo/query/acc.cgi?acc=GSE193105 | NCBI Gene Expression Omnibus, GSE193105 |

The following previously published dataset was used:

| Author(s) | Year | Dataset title | Dataset URL | Database and Identifier |
|---|---|---|---|---|
| Erkosar B, van der Meer JR, Kawecki TJ | 2017 | RNAseq on *Drosophila* larvae genetically adapted to poor diet in microbiota-colonized and germ-free state | https://www.ncbi.nlm.nih.gov/bioproject/PRJNA412704 | NCBI BioProject, PRJNA412704 |

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
