## [Editor Report]

This important study pushes forward the understanding of a major question in evolutionary biology and human health: does juvenile malnutrition affect the performance of the adult? Combining experimental evolution in *Drosophila* with transcriptomics, metabolomics, and genomics datasets, it provides solid and compelling evidence that adult adaptation is constrained by adaptation at the larval stage. The work will be of interest to a broad audience including evolutionary biologists as well as human health researchers.

---

## [Decision Letter]

**Decision letter after peer review:**

[Editors’ note: the authors submitted for reconsideration following the decision after peer review. What follows is the decision letter after the first round of review.]

Thank you for submitting the paper "Evolution under juvenile malnutrition impacts adult metabolism and fitness in *Drosophila*" for consideration by *eLife*. Your article has been reviewed by 3 peer reviewers, one of whom is a member of our Board of Reviewing Editors, and the evaluation has been overseen by Detlef Weigel as Senior Editor. The reviewers have opted to remain anonymous.

We are sorry to say that, after consultation with the reviewers, we have decided that in its current state, this work will not be considered for publication by *eLife*. However, the reviewers agreed that the work represents an extensive collection of datasets that addresses a very timely and important question in biology. We will therefore aim to contact the same editors and reviewers if you decide to make a new submission with a thoroughly revised manuscript.

Specifically, there was consensus in that the manuscript should (a) better contextualize the motivation/results/conclusions using previous data/theory, (b) tone down the overinterpretation of results, and (c) state clearly the biological aspect that is being tested with each experiment and argue whether the experiments/data used for each question are indeed appropriate.

We believe the reviews were very thorough and will help improve the manuscript. We therefore attach them below in their entirety.

*Reviewer #1 (Recommendations for the authors):*

In this paper, Erkosar et al. explore a fundamental question in evolutionary biology as well as in human health: what are the consequences of juvenile malnutrition for adult performance. They use *Drosophila melanogaster*, a holometabolous insect, as a study system. By undergoing full metamorphosis, it could be assumed that larval and adult stages have been decoupled in this species. This system therefore allows the authors not only to address a general question regarding juvenile vs adult effects, but also a more specific one regarding the uncoupling of larval and adult stages in this type of insect. A strength of this paper is the use of an experimental evolution set up with 6 replicates each for the selection and control regimes – this allowed the authors not only to study the effects of juvenile malnutrition within a generation but also the long-term effect in adults of 230 generations of larval malnutrition. Previous studies from the authors and others had characterized certain higher-order adult phenotypes (lifespan, fecundity, starvation, etc) in this and other selected populations, and now, Erkosar et al. gather metabolomics and transcriptomics data to connect genetic variation (from a previous study) to expression changes to metabolism to higher-order phenotypes like fecundity and starvation resistance. This type of complete dataset crossing different levels of organization in an organism is not common, and the authors deserve praise for that.

The paper makes two main points that are extremely interesting but should be taken with some caution:

First, selection on larvae malnutrition results in low-fitness adult phenotypes. Using a very complete dataset that combines genomics, transcriptomics, metabolomics, and higher-order phenotyping, it is shown that at all levels, the main physiological change underlying low fecundity and low starvation resistance is purine and amino acid metabolism. The big-picture question here is why adaptation to juvenile malnutrition should constrain adaptation in the adult stages in a holometabolous species. To explain this, the authors argue that larva and adult transcriptional profiles are indeed not decoupled, they are strongly correlated. And therefore, adaptive changes at the larval stage are carried over to the adult stage where they are not adaptive. Although the first part is based on solid data, the second part might not be:

1. The authors show that overall, the expression levels are correlated between larvae and adults (r = 0.42) and that most of the differentially expressed genes have the same sign in both stages. However, the genes that are associated with purine and amino acid metabolism in adults, which the authors argue underlie low starvation resistance and fecundity, are not differentially expressed in larvae. This questions whether the coupling argument can really explain the low-fitness phenotypes in the adult, given that data, it seems that this cannot be concluded. The authors should clarify.

2. In the methods section, the dataset for larvae gene expression used to argue the point above (Erkosar et al. 2017) is said to have been collected for larvae which "were raised on the poor diet". This means that the differential gene expression in that dataset corresponds to the plastic response of larvae to poor diet (i.e., selected larvae on poor diet vs control larvae on a poor diet) and not to the selected response (i.e., selected larvae on poor diet vs control larvae on a standard diet). If this is true, then the correlation estimated in the paper corresponds to plastic response in larvae vs selected response in adults, which will further prevent us from concluding that the expression profile of selected larvae and adults are coupled. The authors should clarify the dataset that was used for this comparison, and how this can change the interpretation of the results.

The second big (in my opinion) conclusion of the study is that plastic response to larvae malnutrition is independent of the selection response to the same stimuli. This sounds counterintuitive but is supported by two independent datasets, gene expression and metabolomics, which suggests it is a solid result. Despite such an interesting pattern, the authors do not really digest this result for the reader, nor contextualize it in any way, despite being other studies exploring the relationship between plastic and adaptive response (e.g. in fish Ghalambor et al. 2015, in flies Huang and Agrawal 2016 Plos Gen, in plants Josephs et al. 2021 Evol letters, etc). The data, nor the discussion of it, help connect the dots: there is a correlation between larval expression profile and adult selected profile, but no correlation with the intermediate step between these two which is the plastic response in the adult?

So, I am left wondering whether conceptually the authors are asking the wrong question here. Given that adults never experience any change in diet, they cannot possibly be addressing plasticity with the experimental design used here? Maybe actual plasticity can only be measured in larvae that experience both diets? Or maybe everything makes sense? In any case I think the authors should discuss better this result for it to be meaningful.

1. It seems like the 'coupling' of larvae and adult expression profiles is a surprising result. However, despite two lines here and there, this is not really contextualized in the text. For someone not working on insects, there is no way of weighing this result. The authors should improve the description of (a) what are the expectations regarding coupling/decoupling and what is previously known about this, (b) the biology of metamorphosis. For example, there are certain things that are "locked" during larval development like size (as accounted for during the ovarioles analysis) and therefore larva and adult phenotypes are coupled. What other things are "locked" during larval development and how can they explain the 'coupled' patterns of gene expression you see? E.g. are some of those pathways things that we should expect to be decoupled/coupled?

2. The result section should be re-organized to bring together data that addresses the same question. For example, the first section 'gene expression patterns point to evolutionary changes in metabolism" uses half of the space describing the comparison selection regime vs plasticity and only at the end addresses the evolutionary changes in metabolism. And then, the second section does discuss the overlap of plasticity and evolution regime but half of this comparison had been described in the first section. This happens again in other result sections and makes the paper harder to read (e.g. pages 7 and 9).

3. Page 4. You say the first PCs don't separate the groups and the PCA is not shown. Could you please add this as a SI figure, and state what is driving those first 3PCs before the signal you are interested in pops up in PC4-5.

4. Page 6, last line. It cites Figure 2 but that PCA is not shown in that figure or anywhere else. This figure will be interesting to put next to the PCA for expression (point 4 above) given that the expression signal is weaker than the metabolomics in separating treatments. Also, you should discuss why this is so in the discussion.

5. First line of page 5. I think it is standard vs poor diet instead of selected and control?

6. Parts of the methods are described in the Results sections, which is OK, but then, that information is omitted from the methods. Please add all relevant info to the methods section.

7. Just a suggestion of course. I think the title does not do justice to the article. It is already known that juvenile malnutrition impacts adult phenotypes. But you went well beyond that! What about something like "evolution under juvenile malnutrition show that larval and adult stages are not decoupled in *D. melanogaster* resulting in low-fitness adults".

In general, the paper will benefit a lot from better visualization of the results. There is too much text that could go into plots:

8. It will help the reader a lot to have a diagram of your experimental setup. It's hard to imagine plastic response vs selection because the selection regime also experiences plastic response every generation when adults are placed in standard food. You should make a drawing of the life stages of *Drosophila* and to which diet each one is exposed, and highlight the differences between selection regimes (230 generations, etc) and the plasticity experiment.

9. The overlap between selection and plasticity for gene expression and metabolomics could be made into a plot that shows the number of genes that overlap and the direction of change. Something like x-axis is selection lfc and y-axis is plasticity lfc.

10. Figure 1 could have more information given the first two sections of the results. Panel A is OK, Panel B, instead of showing up and down-regulated for selection regime could instead compare GO terms between selection regime and plasticity – again to highlight that nothing overlaps except for cuticle GO. It will be useful to see the p-values of the enrichment. The plot mentioned in 8 could go as a panel here.

11. Page 6 first section. This could also be made into a plot comparing lfc in larva and adults. And also add the comparison of the GO terms because here there are interesting differences.

12. Page 7 and 9, The results could be shown in plots of PCA, the comparison of lfc between selection and plasticity, and GO terms.

*Reviewer #2 (Recommendations for the authors):*

Nutritional stress early in life is pervasive and has a wide range of negative effects on adult physiology. Whether the genes underlying the physiological response that allows organisms to overcome stress early in life also enable adult success remains poorly understood. The evolutionary response may favor plastic strategies, where flexibility in response to different stress increases under early-life stressors. Alternatively, genetic variants that enable larval survival may have pleiotropic effects that negatively impact adult fitness. Untangling whether these effects are independent or interactive requires clever experimental design in a tractable system.

Here, the authors use experimental evolution in *Drosophila melanogaster* to understand evolution to poor nutrition early in life. For >200 generations, larvae were reared in a nutrient-poor diet (SEL diet, 1/4 the concentration of the standard diet). Then, after eclosion, flies are reared under a standard diet. The control (CTL) population was reared on the standard diet for the entire lifespan. For the results presented here, Erkosar et al. performed a reciprocal transplant experiment, where both the control (CTL) and selected (SEL) flies were reared in both CTL and SEL diets. This enabled comparisons between the effects of the evolutionary regime (CTL and SEL flies) and plasticity by varying the within-generation dietary effects (CTL and SEL diet). Combining RNAseq, metabolomics, and genomics with experimental evolution provided a comprehensive measure of how the evolutionary response to dietary stress shaped fly physiology.

A key strength of the study is the experimental design and comprehensive phenotyping. By rearing SEL and CTL lines in both SEL and CTL dietary regimes, the effects of evolution and plasticity can be better understood. In general, the RNAseq and metabolome results suggest that the evolutionary regime and dietary regime act additively in shaping adult physiology, but increased plasticity did not evolve (as no significant interactions were observed). One concern is that gene expression analyses was limited to the adult fat body, where the top hits were primarily associated with cuticular development (SEL flies are smaller). Additional tissues or at different time points in development might provide clearer insights into how evolution under dietary stress shapes gene expression and metabolism. Metabolomics also highlighted the role of purine metabolism. Purine metabolism was upregulated in SEL flies, which were also associated with SNPs that differentiate the genomic response between SEL and CTL lines.

A particularly interesting experiment was to starve the flies and profile the resulting changes to the metabolome. The SEL lines reared on the CTL diet still had a metabolome that more closely resembled the CTL fly after starvation, suggesting that the SEL flies exist in a permanently starved state, regardless of the current diet. The consequences of this permanently starved state also impact fecundity. SEL flies had lower fecundity than CTL flies on the SEL diet, counter to expectations that SEL lines would show greater fecundity in the SEL diet. SEL flies are thus paying a fitness cost for the traits that enabled survival in nutrient-poor diets. However, fecundity was only assayed in the SEL diet, and so it is unknown whether the reduction in fecundity for SEL flies is in response to selection or the diet.

Overall, the authors propose that the results support a scenario where the cross-life stage, pleiotropy drove adaptation to the dietary stress. In other words, SEL flies pay a fitness cost late in life for the strategies used to survive the nutritional stress early in life, with signatures of selection at the genomic, transcriptomic, and metabolomic levels. Particularly for organisms like *Drosophila* that inhabit ephemeral and fluctuating environments, plasticity is thought to evolve to decouple the physiological needs that differ between adults and juveniles. However, here, through experimental evolution and 'omics rich data, Eroksar et al. provide new insights into the role of pleiotropy in shaping adaptation to early life stress.

I really appreciated the multifaceted approach to understanding adaptation to nutritional stress early in life. The experiments are necessarily complex, and the manuscript is very rich in data-but, I had a difficult time throughout keeping track of how the different experiments linked to different results/conclusions/implications, etc.

One issue is that plasticity wasn't well defined to begin in the introduction. The authors begin at line 46 with the statement "In contrast to such phenotypically plastic within-generation responses", but is not clear how the first paragraph showed phenotypic plasticity. The second paragraph suggests that one can examine plasticity by placing larvae in a different environment than adults. However, this is also the selection regime, where only larvae are reared on nutrient-poor diets and adults are reared on the same standard diet. Plasticity can be assessed in different ways (e.g., change in phenotypic variance, intragenotypic differences, reaction norms, etc.), and so a clearer definition would help ground the complexity of the following experiments. Further, does the starvation experiment count as plasticity in this framework? Clarification would help the reader understand and appreciate the complex experiments that are necessary to address this question.

While the description of the five questions (lines 78-103) helps provide some context in how the reviewers are thinking about plasticity, it was disconnected with the subject material in the introduction and introduced further complications like pleiotropy. It might help by having a simpler statement in the introduction describing how plasticity can be assessed using the selected and control populations.

Another general comment is the error bar in Figure 3B, Figure 6B, etc. It is a little unusual to see just one error bar per panel in the figures. I also see that the error bar is the standard error of the mean-but mean of the main effect (evolutionary regime)? I see the placement is zero centered for the log fold change, which is OK, but what is it for Figure 6B? An additional explanation would be appreciated.

Many of the experiments need additional rationale to describe them. While I got this to some extent in the introduction lines 78-103, the rationale and implications are underdeveloped for several of the key results.

In the experiment associated with Figure 1, why are fat-body/carcasses used? It is described that the fat body is the key metabolic organ, but why wouldn't the whole body be used? For example, given the fecundity results presented later, how does the fat body transcriptome inform these results as opposed to what ovary or whole-body results would show? What is lost by focusing just on this one tissue group? If cuticular development and maturation genes are still most important, wouldn't that still show whole body RNAseq? Was the choice to enable linking the RNAseq with the metabolomics that follows? The rationale should be better clarified.

I'm also not sure I understood also why only the evolutionary regime differential expression results were shown, but not the larval diet results (summarized lines 162-188). I found the lack of parallelism an interesting result from the RNAseq experiment and would find them useful visualized (not as a table) at least in the supplement.

The comparison with Erkosar et al. 2017 was not well justified. In the methods, lines 614-15 it is mentioned that the larval gene expression is derived from an experiment that manipulated the microbiome, with and without microbes. The evolutionary regime was used as the main effect for comparison, but this isn't totally straightforward-why not just subset the 2017 data for only +microbe? why include the germfree at all here? Also, Erkosar 2017 data is from generations 177-200, which isn't stated here. It's very surprising that the same genes are differentially expressed in mostly the same way between larvae and adult carcasses given all these differences in experimental design. The differences between these datasets should be explicit for the reader. However, given the complications listed above, I feel like the comparison between the carcass data and the Erkosar data does not add much to the overall message (and it's not visualized).

I liked the starvation experiment. Like before, it needs to be clear whether plasticity here refers to the std-poor diet and/or fed-starved comparison. It can be both but needed more information here. I think the authors consider the results to show more about the evolutionary regime (lines 373-375), but why isn't this also an aspect of plastic responses to nutritional stress? The interactions between fed-starved and the connection with differential allocation are very interesting results. While most of the discussion of this result is focused on the SEL resembling a starved metabolome, the change for SEL between fed and starved seems less than for the CTL. The CTL-poor seems to change most dramatically between fed and starved, which makes sense given the previous results of this selection experiment. Does this mean that SEL can buffer the nutritional stress better? I'm curious how buffering of this starvation stressor, despite previous evidence that SEL are less starvation resistant (line 331), informs our understanding of the adaptation to nutritionally poor larval diets.

Great to see some overlap between genomic, transcriptomic, and metabolomics in purine synthesis. Please make clear that the Kawecki 2021 genomic data is from ~150 generations.

For the adult costs of larval adaptation, it is interesting that CTL laid more eggs than SEL populations when exposed to a poor larval diet. Would you see the same result if fecundity was assayed for both on the standard diet? i.e., does the evolutionary regime just lead to generally fewer eggs produced by SEL populations? Does it matter for adaptation to nutritionally poor diets if SEL populations are just less fecund? I think the rationale needs some additional justification here. Additionally, in Figure 5, can these be visualized in a way that shows the true range of observed values (e.g., boxplots)?

I think the discussion provides important context, and the authors highlight one potential explanation for the results is pleiotropy across life stages. This is an interesting perspective given that *Drosophila* are holometabolous insects, and as the authors say, this remains surprisingly understudied. It's not clear whether this cross-life-stage pleiotropy is amplified because of holometabolous insects or also generalizable to other organisms as well.

*Reviewer #3 (Recommendations for the authors):*

Erkosar et al. use gene expression and metabolomic analyses to investigate the evolutionary change that accompanied over 230 generations of selection by nutrient deprivation from egg to pupation in *Drosophila*.

The evolved populations were six replicate populations, each selected for larval development and growth under poor nutrient conditions. Each population was kept at ~200 flies at each generation, which may have allowed significant drift when compared to larger populations in similar experimental evolutionary studies however, this regime has resulted in repeated evolution of a few interesting traits. These traits were revealed in earlier work and include faster developmental time, higher egg-to-adult viability, and smaller adult size. Additionally, selected lines showed higher sensitivity to starvation as adults and the allele frequency changes that accompanied selection intersect with those loci associated with selection for starvation resistance.

The picture painted from previous work is of genotypes arising from selection that are more likely to make pupae/adults; to make them faster and more numerously than the control population. The adults that they make however are not as healthy as the adults from the control population. This certainly suggests that adult traits suffer a constraint based on the evolution of juvenile condition. Is that constraint manifest in patterns of gene expression, or in patterns of variation in the metabolome? Hints at metabolic adaptation within these populations came from a prior publication that described a difference in nitrogen or carbon assimilation between these populations, which is at least intriguing.

The current work is important in that it sheds light on the kinds of endophenotypic changes that accompany these evolutionary changes. In addition to the condition-specific patterns of expression and metabolite abundance, the authors also test their ideas about how the transcriptome and metabolome of adults of both populations respond to diet, and if there is a resemblance to flies evolved on poor food and starved flies.

This paper presents several interesting results. One, the evolution that occurred in the larval transcriptome was largely similar to the differences between the populations as adults. This implies a constraint in adult phenotypes on juvenile phenotypes and vice versa that is reflected in the transcriptome.

Within the same experiment, the transcriptome and metabolome of adults from both populations appear to respond similarly to diet. That is, the adults of both populations, despite significant shifts in metabolome and transcriptome, show very similar responses to being placed on the poor diet, when compared to the normal diet. This is an unexpected result given the constraint that may explain the previous result.

That the within-generation response to diet appears independent of the evolutionary response to diet conditions is interesting. The authors could develop their discussion of these results further. I think it is worth considering that the effects of diet with a generation are not necessarily adaptive. An adult on a poor diet might have a metabolome that looks a certain way, reflecting in part the exhaustion of resources. The adult from a larva that has evolved on a poor diet however may more closely approximate an adapted metabolome, there is really no reason to think that these would be the same.

It is also interesting that the effect of diet on the adult metabolome was somewhat similar to the difference between the adult metabolome of the two populations, the evolutionary change. This is in contrast to the seemingly independent effect of diet on the transcriptome, and the transcripts that differ between the populations. I hope that this result is not a 'false positive' due to the small biological sample size of the transcriptome data (n=1) relative to the metabolome data (n=2), or a 'false negative' due to the relatively sparse metabolome coverage (~100 metabolites) that is typical of targeted analysis of fly metabolome compared to the more comprehensive transcriptome. This result is in line with other studies that indicate that endophenotype appears to converge as it goes from: gene > RNA > protein > metabolite, to phenotype. The relative anticipation of the evolutionary response seen in the metabolome differences by diet when compared to the same in a relationship in the transcriptome is neat.

The results described above manifest largely in PCA, in reduced dimensional space, which is appropriate and telling, but may hide variation that occurs within and between populations, and across diets. There is some agreement within the univariate analysis of metabolites and transcripts that are differentially abundant between populations and treatments, suggesting that the 'biological signal' seen in the PCA could be picked apart using the identities of the genes and metabolites. The authors indulge in this pursuit and while creative, take their interpretation too far. The authors make several clever interpretations of their data in order to shed light on potential mechanisms (pathway activity) that may explain their results. Without testing some of these ideas however this amounts to speculation.

The third point of interest is the "Metabolic profile of Selected flies tends towards starved-like state", an argument that the metabolome of selected flies more closely resembles starved flies than does the metabolome of control flies. This is argument rests on MANOVA and some agreement in sign with metabolite levels upon starvation, and those affected by selection. This is an interesting result that could relate to the sensitization to adult starvation in the evolved population. At the univariate level however, the picture is quite complicated (Figure 3B). The interactions between diet, selection regime and the effect of starvation are complex. The authors make a valiant effort at interpretation, however, I don't think their conclusions are well supported from these data. I suggest approaching the univariate analysis more circumspectly.

The authors are over-interpreting the meaning of their data. This is my main objection to what is otherwise an interesting and meaningful study. I think that the authors should include their interpretation of the data, however, it should be presented as speculation, rather than as if the interpretation were itself an observation.

The joint analysis presented in 'Figure 4—figure supplement 1' is problematic for several reasons. I don't think it should be included as it does not add anything to the paper worth discussing.

"[abstract line # 26]: "resulting in deficiency of electron transports and congestion in β-oxidation." Two objections, first, 'deficiency' is different from 'reduced', we have no idea if the reduced level of these RNAs is at all affecting the trait, these are associations and cause and effect is not known. Second, RNA is not protein, and even then, protein abundance is not activity, so either discuss these results in a more speculative light or make a more direct and independent test of electron transport activity/function. Further, congestion in β oxidation is certainly a possible explanation for the patterns of acetylcarnitines and other FA-derivatives that are more/less abundant, however, this too is a speculative explanation and ignores the equally valid evidence that this explanation is wrong.

Similarly, the section that discusses the amino acids [#492-505] describes some amino acids as 'deficient' in the selected flies. These amino acids are reduced in abundance compared to the control flies, but this is not the same as deficient. Deficiency implies that if there were more of these then the outcome would change. That result is unknown. The data are describing an association with abundance, concluding that this is a deficiency is misleading.

In line #385, "This analysis confirmed that the changes in amino acid and purine metabolism contributed to adaptation to poor larval diet." Confirming that something contributes, implies that you have gone beyond association and tested causality. These data do not demonstrate cause or effect but bolster a previously-identified association.

There are two sections of the results (lines 217-290, and 378-404) dedicated to the pathways that might be involved in the transcriptome and metabolome differences seen in these populations. I suggest paring this down, at least as a part of the Results section, as enough of it is speculation. Lines #378-404 are mostly speculative and so this argument should move to the discussion. Similarly, the discussion could be pared down given some of the contradictions that emerge when the data are discussed. For instance, if the paragraphs from lines #492-505 resolves with the axiom: "the differences in amino acid abundance between Selected and Control populations is consistent with differential use of amino acids in metabolism – their interconversion, catabolism and use in the synthesis of other metabolites.", then I'm not sure that it's a good use of space.

Figure 6 could be idealized (made with generic genes) or excluded. I say this because there are at least 10,000 genes from which to pull patterns like these from, so seeing these genes like this does not have meaning (FDR is high). The idea behind this figure can certainly be discussed as it is in the discussion, but this data set is underpowered for this kind of interference.

The authors use the term pleiotropic at several points. I think they are referring to something like antagonistic pleiotropy. I think a short section that introduces this term in this way, with reference to the theoretical or experimental work on the idea in relation to the evolution of aging perhaps would help the reader.

In the bigger picture, there are 'inward' and 'outward' implications for this work. The authors present support for their conclusion that negative trade-offs in adulthood are tied to the endophenotypic changes that occur in juveniles as a result of selection when only applied to juveniles. This support comes from the within-population similarity in adult transcriptome and metabolome to that of larvae. The 'inward' perspective, while tantalizing, is perhaps over-presented in the discussion of these results. Insight into the mechanisms (cellular pathways) by which these trade-offs happen can and should be included, however there are also wider implications of these results for insect evolution, the 'outward' perspective. These results suggest that the evolution of juvenile phenotypes is constrained by the outcomes they associate within the adult, at least on juvenile nutrition inputs/needs. The authors cite some theoretical papers on the topic, and I think the reader would be better served if the authors could reflect on the implications of the constraint they appear to have found. What does this imply for the phenotypic (and environmental) space that the juveniles are constrained to? We know that *Drosophila* are very choosy when it comes to oviposition, and here we see one of the reasons why that might be the case. Do the authors think that adult choice has evolved in part to deal with this constraint? It's easy to see the negative effects, the burden, of such a constraint. Evolution is powerful though, so why then is such constraint 'allowed'? Could this constraint actually reflect some kind of as-yet unconsidered benefit? For instance, does a constrained gene-trait map allow *Drosophila* to maintain a smaller, simpler, genome? I am not familiar with the literature on this and other topics that come to mind when the weight of these results are realized. Perhaps the authors could spend some of the discussion to share their interpretation?

There are a few passages that I found hard to understand. When describing part of their hypotheses about the evolution of adult and juvenile phenotypes, they write [#82-85]: "Plastic responses that program metabolism adaptively for a poor nutritional environment in adulthood would be misdirected in the Selected populations because the adults were switched to standard diet; evolution should thus counteract them." Why would plastic responses in adults be affected by selection in juveniles? If they were affected, why would we call them plastic?

Keep in mind that this sentence is followed by [#85-87]: "Conversely, plastic responses that alleviate the consequences of having developed on a poor diet irrespective of adult diet should have become amplified as a result of evolution on the poor diet." It sounds like the authors are predicting that two counteracting effects have both happened and this does not make sense to me. I would like to understand the authors' point here and I would hope that it could be clarified in a revision.

There were several references to the figures that don't seem correct, and other parts of the figure referencing scheme that did not make sense. For example, the reference to 'Figure 2' on line #231, refers to a PC plot, and not the heatmap shown in Figure 2. Line #231 refers to Supplementary Figure file 4, and I'm not sure that's correct. Please make sure that the figure references point the reader to the correct figure and that the figures are referred to and arranged in numerical order.

The experimental design, the sampling scheme and batch design of the analyses were all good. These experiments made the most of the number of samples that were used. The statistical analysis as described is appropriate, however there is enough going on that making the code available would probably help raise the impact of this work.

The authors are clearly aware of the multiple testing problem and I stand by my recommendation to remove figures that make use of data that do not survive FDR correction.

A point I will need clarification on occurs in lines [#601-604]: 'Therefore, we also estimated the number of genes that differed in expression due to each factor in the analysis as the total number of genes minus the estimated number of "true nulls" (Storey and Tibshirani 2003) (https://bioconductor.org/packages/topGO/), using the weight method and Fisher's exact test.' What does this mean? By 'factor in the analysis', I assume you mean regime and diet. Why, and how, would you use topGO to evaluate the number of genes affected by regime or diet?

---

## [Author Response]

[Editors’ note: the authors resubmitted a revised version of the paper for consideration. What follows is the authors’ response to the first round of review.]

Reviewer #1 (Recommendations for the authors):In this paper, Erkosar et al. explore a fundamental question in evolutionary biology as well as in human health: what are the consequences of juvenile malnutrition for adult performance. They use *Drosophila melanogaster*, a holometabolous insect, as a study system. By undergoing full metamorphosis, it could be assumed that larval and adult stages have been decoupled in this species. This system therefore allows the authors not only to address a general question regarding juvenile vs adult effects, but also a more specific one regarding the uncoupling of larval and adult stages in this type of insect. A strength of this paper is the use of an experimental evolution set up with 6 replicates each for the selection and control regimes – this allowed the authors not only to study the effects of juvenile malnutrition within a generation but also the long-term effect in adults of 230 generations of larval malnutrition. Previous studies from the authors and others had characterized certain higher-order adult phenotypes (lifespan, fecundity, starvation, etc) in this and other selected populations, and now, Erkosar et al. gather metabolomics and transcriptomics data to connect genetic variation (from a previous study) to expression changes to metabolism to higher-order phenotypes like fecundity and starvation resistance. This type of complete dataset crossing different levels of organization in an organism is not common, and the authors deserve praise for that.

We greatly appreciate this praise from the reviewer. To strengthen the evidence for evolutionary non-independence of larval and adult physiology, we now added a comparison with a new data set from larval metabolomics. These data and their analysis from the viewpoint of larval adaptation are published elsewhere (https://doi.org/10.1093/evlett/qrad018), but we used it here to test if we see a similar correlation between the effects of genetically-based evolutionary change on larval and adult metabolite abundance as we see for gene expression. We do (Figure 5B). Furthermore, with this new data set we could also ask if the phenotypically plastic responses of larvae and adults to larval diet are correlated – and they are not (Figure 5 —figure supplement 1). This implies that differences in adult metabolite abundance cannot be explained by some kind of general metabolic "inertia", where difference accrued during larval stage persist into adult stage. I.e., the metabolome differences between Selected and Control populations are more likely to be mediated by the expression of differentiated genetic variants during the adult stage rather than by lingering consequences of expression of these variants during the larval stage.

We believe these new analyses strengthen the case for the interpretation that a large fraction of genetic variants favored by selection on poor larval diet affect larval and adult metabolism similarly, constraining their independent evolution.

The paper makes two main points that are extremely interesting but should be taken with some caution:First, selection on larvae malnutrition results in low-fitness adult phenotypes. Using a very complete dataset that combines genomics, transcriptomics, metabolomics, and higher-order phenotyping, it is shown that at all levels, the main physiological change underlying low fecundity and low starvation resistance is purine and amino acid metabolism. The big-picture question here is why adaptation to juvenile malnutrition should constrain adaptation in the adult stages in a holometabolous species. To explain this, the authors argue that larva and adult transcriptional profiles are indeed not decoupled, they are strongly correlated. And therefore, adaptive changes at the larval stage are carried over to the adult stage where they are not adaptive. Although the first part is based on solid data, the second part might not be:1. The authors show that overall, the expression levels are correlated between larvae and adults (r = 0.42) and that most of the differentially expressed genes have the same sign in both stages. However, the genes that are associated with purine and amino acid metabolism in adults, which the authors argue underlie low starvation resistance and fecundity, are not differentially expressed in larvae. This questions whether the coupling argument can really explain the low-fitness phenotypes in the adult, given that data, it seems that this cannot be concluded. The authors should clarify.

We reported in the previous version that the genes that were differentially expressed between SEL and CTL in larvae were not enriched in amino acid and purine metabolism GO terms.

Prompted by the reviewer's comment, we had a closer look at genes involved in amino acid and purine metabolism in the larval RNAseq data. In fact, we found that 18 out of 99 genes in GO "α amino acid metabolic process" and 16 out of 107 in GO "purine-containing compound biosynthetic process" were significantly different between Selected and Control larvae at 10% FDR. These numbers are greater than the corresponding numbers for adults (9 out of 95 for amino acid metabolism genes, 7 out of 78 for purine biosynthesis). These GO terms are not enriched for the larvae because in general many more differentially expressed genes found in the larvae (21%) than in the adults (2.5%). Hence, while the amino acid metabolism and purine synthesis do not show disproportionate changes in gene expression in the larvae compared to other GO terms, the expression of multiple genes in those pathways has clearly been affected. Furthermore, although very few of those genes pass the 10% FDR in both larvae and adults, the differences between SEL and CTL across all genes in those GO terms are positively correlated between larvae and adults (amino acid metabolism: r = 0.40, P < 0.0001, N = 94; purine compound synthesis: r = 0.47, P < 0.0001, N = 74). In contrast, no such correlation is found for the top GO term for adults "chitin-based cuticle synthesis" (r = 0.05, P = 0.65, N = 107).

We had overlooked this in the first submitted version, and we are grateful to the reviewer for their comment that made us re-examine these results. We have now modified the corresponding paragraph of the Results section (l. 252-261) to include the above observations.

2. In the methods section, the dataset for larvae gene expression used to argue the point above (Erkosar et al. 2017) is said to have been collected for larvae which "were raised on the poor diet". This means that the differential gene expression in that dataset corresponds to the plastic response of larvae to poor diet (i.e., selected larvae on poor diet vs control larvae on a poor diet) and not to the selected response (i.e., selected larvae on poor diet vs control larvae on a standard diet). If this is true, then the correlation estimated in the paper corresponds to plastic response in larvae vs selected response in adults, which will further prevent us from concluding that the expression profile of selected larvae and adults are coupled. The authors should clarify the dataset that was used for this comparison, and how this can change the interpretation of the results.

We think there has been some misunderstanding here as to what we mean by the evolutionary response and the plastic response. An evolutionary change in a phenotype is, by definition, mediated by changes in the genetic composition of the population, so it must be assessed in the same environment (a "common garden"). Thus, "Selected larvae on poor diet vs Control larvae on a Poor diet" measures an evolved, genetically-based divergence, not a plastic response. Similarly, "Selected adults raised on poor larval diet vs Control larvae raised on poor larval diet" measures the phenotypic divergence caused by these same genetic changes in adult phenotypes.

The contrast "Selected on poor diet vs Control on standard diet" proposed by the reviewer compares different genotypes in different environments; i.e., it combines effects due to evolution and due to plasticity. Thus, even though these are actually the phenotypes exposed to selection, we do not use them for comparisons e.g. across the life stages (furthermore, we do not have larval gene expression data on standard diet).

We now explain more explicitly at the beginning for Results the meaning of phenotypic plasticity and evolutionary response (l. 149-154).

The second big (in my opinion) conclusion of the study is that plastic response to larvae malnutrition is independent of the selection response to the same stimuli. This sounds counterintuitive but is supported by two independent datasets, gene expression and metabolomics, which suggests it is a solid result. Despite such an interesting pattern, the authors do not really digest this result for the reader, nor contextualize it in any way, despite being other studies exploring the relationship between plastic and adaptive response (e.g. in fish Ghalambor et al. 2015, in flies Huang and Agrawal 2016 Plos Gen, in plants Josephs et al. 2021 Evol letters, etc). The data, nor the discussion of it, help connect the dots: there is a correlation between larval expression profile and adult selected profile, but no correlation with the intermediate step between these two which is the plastic response in the adult?

We have now included these references, as well as another one by Yampolsky et al. (2012). We note that three of those four studies found no or a somewhat negative correlation between the plastic and evolutionary responses; only Josephs et al. found a positive correlation. We do not think this result is counterintuitive, it just suggests that the "evolution follows plasticity" hypothesis proposed by some authors does not seem to be generally supported by this kind of data.

Also, we do not see in what sense the plastic response of the adult is an "intermediate step" between a genetically-based evolved change of larval expression patterns and a genetically-based evolved change in adult expression patterns. The most parsimonious explanation for this pattern is that genetic variants that were the raw material for the evolutionary divergence between Selected and Control populations tend to affect expression at both larval and adult stage, and in the same direction. We now elaborate on the background of this issue in the second and third paragraph of Introduction and further discuss the interpretation of our results in Discussion.

So, I am left wondering whether conceptually the authors are asking the wrong question here. Given that adults never experience any change in diet, they cannot possibly be addressing plasticity with the experimental design used here? Maybe actual plasticity can only be measured in larvae that experience both diets? Or maybe everything makes sense? In any case I think the authors should discuss better this result for it to be meaningful.

Phenotypic plasticity, according to a widely accepted definition, is the expression of different values of phenotypic traits by the same genotype(s) in response to different environmental conditions. This includes cases where the environmental factor of interest is acting at an earlier life stage than the stage when the phenotype is measured. Indeed, as we state in the Introduction, there is a lot of interest in the plastic responses of adult metabolism to juvenile or developmental nutrition, also in the context of human health. Thus, we argue that asking whether this plasticity of adult physiology in response to larval diet anticipates the evolutionary change of the same adult phenotypes driven to the same larval diet is a legitimate and interesting question.

However, the reviewer is right that the fact that the evolutionary change has been driven by larval diet and we are focusing on adult metabolic phenotypes renders the interpretation of the relationship between the plastic and evolutionary change more difficult. This is in part because, as our results suggest, the evolutionary changes of the adult phenotype may be maladaptive. We now elaborate on this in the Discussion (l. 253-272)

Obviously, plasticity of the larvae in response to larval diet and its relationship with the evolutionary change in larval phenotype is an interesting question, which we have addressed in another paper (https://doi.org/10.1093/evlett/qrad018). We found no such relationship.

1. It seems like the 'coupling' of larvae and adult expression profiles is a surprising result. However, despite two lines here and there, this is not really contextualized in the text. For someone not working on insects, there is no way of weighing this result. The authors should improve the description of (a) what are the expectations regarding coupling/decoupling and what is previously known about this, (b) the biology of metamorphosis. For example, there are certain things that are "locked" during larval development like size (as accounted for during the ovarioles analysis) and therefore larva and adult phenotypes are coupled. What other things are "locked" during larval development and how can they explain the 'coupled' patterns of gene expression you see? E.g. are some of those pathways things that we should expect to be decoupled/coupled?

We have now elaborated on the rationale and interpretation of this cross-stage coupling in two places. First, we are bringing up the notion of metamorphosis "decoupling" larval and adult phenotypes in the 3rd paragraph of Introduction. Second, in the discussion we very briefly summarize what happens during the metamorphosis, and specifically describe the fate of larval fat body (cell dissociation, autophagy and apoptosis, with possibly some surviving cells contributing, together with progenitor cells, to adult fat body). While obviously the topic is highly interesting, we did not discuss further on what other organs may carry an "imprint" of larval conditions and how, as we felt this would go too much on a tangent. But a new analysis (Figure 5 —figure supplement 1), which indicates an absence of correlation of the plastic response of metabolome to diet between larvae and adults, implies that such imprint does not play a major role in determining adult metabolism.

2. The result section should be re-organized to bring together data that addresses the same question. For example, the first section 'gene expression patterns point to evolutionary changes in metabolism" uses half of the space describing the comparison selection regime vs plasticity and only at the end addresses the evolutionary changes in metabolism. And then, the second section does discuss the overlap of plasticity and evolution regime but half of this comparison had been described in the first section. This happens again in other result sections and makes the paper harder to read (e.g. pages 7 and 9).

We have restructured the Results section in that we put together the relationship between plasticity and evolution for both gene expression and metabolome together in one subsection (l. 274ff), followed by a subsection that describes the correlation in both gene expression and metabolome between larvae and adult (l. 327ff).

3. Page 4. You say the first PCs don't separate the groups and the PCA is not shown. Could you please add this as a SI figure, and state what is driving those first 3PCs before the signal you are interested in pops up in PC4-5.

We now included a figure supplement with plots of the first six PCs (Figure 1 supplement 1). As far as we can say, the first three PCs seem driven by idiosyncrasies of individual populations and samples. Part of this is measurement variance – including e.g. effects of vial (including that microbiome shared by flies) or drift in the dissection procedure (while samples were dissected in a haphazard order, for practical reasons flies from a given sample were dissected together). However, a part is likely to reflect genetic differentiation among replicate populations, which is expected after 250 generations of independent evolution under rather small populations sizes. This is supported by the fact that the PC scores for each of the first 6 PCs of flies raised on standard and poor food are positively correlated across the 12 populations (r = 0.24 to 0.87). As we cannot really say more about this, we simply added the statement "the first three PC axes appear driven by idiosyncratic variation among replicate populations and individual samples."

4. Page 6, last line. It cites Figure 2 but that PCA is not shown in that figure or anywhere else.

This figure was mis-referenced in the previous version (apologies); now it is figure 2A.

This figure will be interesting to put next to the PCA for expression (point 4 above) given that the expression signal is weaker than the metabolomics in separating treatments. Also, you should discuss why this is so in the discussion.

As for the signal of differentiation between treatments (i.e., evolutionary regime and larval diet) being weaker in gene expression than in metabolome, if we could demonstrate this confidently, this would be an interesting observation worth discussing. However, we do not think the PCA provides sufficient support for this kind of statement. While the PCA represents the data in a standardized way, the underlying data are still of a very different nature and thus differentially subject to random biological variation or measurement error. Furthermore, for RNAseq we only had one sample per population and diet whereas for metabolome we had two (not counting the starved fly data). As indicated in the Methods and the figure legend, the duplicated metabolome samples were averaged for the PCA on metabolome. For these reasons, we think any interpretation of the comparison of PCA results for gene expression versus metabolome would be shaky.

5. First line of page 5. I think it is standard vs poor diet instead of selected and control?

The comparison we are making here is indeed between standard and poor diet, but we wanted to state that the top GO term is the same as for Selected vs. Control. We modified the sentence to say (l.276-8):

"As was the case for Selected versus Control populations, the top GO term enriched for genes differentially expressed in flies raised on the poor versus standard larval diet treatment (corresponding to phenotypic plasticity) was 'chitin-based cuticle development' (Figure 1D; Supplementary file 2 table B)."

6. Parts of the methods are described in the Results sections, which is OK, but then, that information is omitted from the methods. Please add all relevant info to the methods section.

We now added several bits of information to the Methods. However, generally we aimed to minimize redundancies between Results and Methods, with the idea that the design and the essence of our methods should be understood from Results and only readers interested in technical details would refer to Methods.

7. Just a suggestion of course. I think the title does not do justice to the article. It is already known that juvenile malnutrition impacts adult phenotypes. But you went well beyond that! What about something like "evolution under juvenile malnutrition show that larval and adult stages are not decoupled in *D. melanogaster* resulting in low-fitness adults".

We had a long back-and-forth about this suggestion among the authors. The title proposed by the reviewer exceeded the 120 characters length limit and we did not find a satisfactory way of expressing the same content in 120 characters. Furthermore, we thought this might too strongly focus the reader on a single aspect of our results. Thus, we finally decided for a version resembling the original one, but with an added emphasis on cost to adult fitness.

In general, the paper will benefit a lot from better visualization of the results. There is too much text that could go into plots:8. It will help the reader a lot to have a diagram of your experimental setup. It's hard to imagine plastic response vs selection because the selection regime also experiences plastic response every generation when adults are placed in standard food. You should make a drawing of the life stages of *Drosophila* and to which diet each one is exposed, and highlight the differences between selection regimes (230 generations, etc) and the plasticity experiment.

We have now added such a scheme (Figure 1A).

9. The overlap between selection and plasticity for gene expression and metabolomics could be made into a plot that shows the number of genes that overlap and the direction of change. Something like x-axis is selection lfc and y-axis is plasticity lfc.

We have now added such plots for both gene expression and metabolome (Figure 4 A,B).

10. Figure 1 could have more information given the first two sections of the results. Panel A is OK, Panel B, instead of showing up and down-regulated for selection regime could instead compare GO terms between selection regime and plasticity – again to highlight that nothing overlaps except for cuticle GO. It will be useful to see the p-values of the enrichment.

We could not think about a useful graphic way of directly comparing the GO terms between two factors, especially that the overlap consists of just one GO term. One could in principle think of some kind of scatterplot where each point would represent a GO term, with the P-value or the proportion of DEG for plasticity on one axis and evolution on the other axis. But most of the points in our case would be on the axes, and trying to place the long names of GO terms would render the plot unreadable. A Venn diagram is also rather impractical.

Furthermore, as we explain above, we came to realize that the overlap or otherwise of enriched GO terms is of limited usefulness as a tool to assess the similarity of response. Thus, we downplayed the lack of overlap between GO terms, focusing instead on examining the correlations at the level of individual genes.

Still, we did add a plot summarizing the top enriched GO terms (i.e. those with P < 0.01) for the plastic response, as these are or interest in their own right (Figure 1D). We retained the form of the plot, i.e., a horizontal bar plot indicating the number of significantly up- and downregulated genes in each GO category, as this reveals some (mildly) interesting patterns, and is slightly more informative than providing just the number of DEG or a plain list of GO terms.

The plot mentioned in 8 could go as a panel here.

Done, as mentioned above.

11. Page 6 first section. This could also be made into a plot comparing lfc in larva and adults. And also add the comparison of the GO terms because here there are interesting differences.

We now added such a plots for logFC for both gene expression and metabolome (Figure 5A,B), they nicely illustrate the positive correlation.

As for the GO terms, as explained above, we are not convinced that a comparison of significant GO terms is particularly useful in this context. Notably, even though amino acid and purine metabolism GOs do not come up as enriched in the larvae, at the level of expression of genes in these categories there is still a clear positive correlation (l. 356-8). Furthermore, a GO-term analysis of the larval expression data has already been published in Erkosar et al. 2017, albeit using a previous version of mapping and with P-value adjustment which seems overly conservative by current understanding. Thus, rather than expand on the GO term analysis in the context of larva-adult similarity, we decided to play it down. We think that the new Figure 5 conveys the message in a sufficiently convincing way.

12. Page 7 and 9, The results could be shown in plots of PCA, the comparison of lfc between selection and plasticity, and GO terms.

It is not clear how this comments add to the reviewer's comments 3,9 and 10 above. As we already explained above, we now have the PCA plots and the logFC for plasticity and evolution for both gene expression and metabolome, and a comparison of GO terms for gene expression.

For metabolome, a somewhat analogous approach is pathway enrichment analysis. As we reported already in the previous version, we attempted to use a join pathway analysis to look for a joint signal of enrichment in metabolite and gene expression data, with rather disappointing or even misleading results (the supposed second top enriched pathway, "valine, leucine and isoleucine biosynthesis" does not exist in animals). We get similar output if we only use metabolite data. A pathway enrichment-based approach seems more suited to untargeted metabolomic data with a large number of features, of which a rather small fraction is differentially abundant. We have 113 metabolites from predefined groups and half of them (57) are differentially abundant. So, even if a particular pathway is not "enriched", it may show substantial changes. For these reasons, we did not elaborate the pathway analyses and did not include their results in a graphical form, just as a supplementary table (Supplementary file 4).

Reviewer #2 (Recommendations for the authors):Nutritional stress early in life is pervasive and has a wide range of negative effects on adult physiology. Whether the genes underlying the physiological response that allows organisms to overcome stress early in life also enable adult success remains poorly understood. The evolutionary response may favor plastic strategies, where flexibility in response to different stress increases under early-life stressors. Alternatively, genetic variants that enable larval survival may have pleiotropic effects that negatively impact adult fitness. Untangling whether these effects are independent or interactive requires clever experimental design in a tractable system.Here, the authors use experimental evolution in *Drosophila melanogaster* to understand evolution to poor nutrition early in life. For >200 generations, larvae were reared in a nutrient-poor diet (SEL diet, 1/4 the concentration of the standard diet). Then, after eclosion, flies are reared under a standard diet. The control (CTL) population was reared on the standard diet for the entire lifespan. For the results presented here, Erkosar et al. performed a reciprocal transplant experiment, where both the control (CTL) and selected (SEL) flies were reared in both CTL and SEL diets. This enabled comparisons between the effects of the evolutionary regime (CTL and SEL flies) and plasticity by varying the within-generation dietary effects (CTL and SEL diet). Combining RNAseq, metabolomics, and genomics with experimental evolution provided a comprehensive measure of how the evolutionary response to dietary stress shaped fly physiology.A key strength of the study is the experimental design and comprehensive phenotyping. By rearing SEL and CTL lines in both SEL and CTL dietary regimes, the effects of evolution and plasticity can be better understood. In general, the RNAseq and metabolome results suggest that the evolutionary regime and dietary regime act additively in shaping adult physiology, but increased plasticity did not evolve (as no significant interactions were observed). One concern is that gene expression analyses was limited to the adult fat body, where the top hits were primarily associated with cuticular development (SEL flies are smaller). Additional tissues or at different time points in development might provide clearer insights into how evolution under dietary stress shapes gene expression and metabolism. Metabolomics also highlighted the role of purine metabolism. Purine metabolism was upregulated in SEL flies, which were also associated with SNPs that differentiate the genomic response between SEL and CTL lines.A particularly interesting experiment was to starve the flies and profile the resulting changes to the metabolome. The SEL lines reared on the CTL diet still had a metabolome that more closely resembled the CTL fly after starvation, suggesting that the SEL flies exist in a permanently starved state, regardless of the current diet. The consequences of this permanently starved state also impact fecundity. SEL flies had lower fecundity than CTL flies on the SEL diet, counter to expectations that SEL lines would show greater fecundity in the SEL diet. SEL flies are thus paying a fitness cost for the traits that enabled survival in nutrient-poor diets. However, fecundity was only assayed in the SEL diet, and so it is unknown whether the reduction in fecundity for SEL flies is in response to selection or the diet.Overall, the authors propose that the results support a scenario where the cross-life stage, pleiotropy drove adaptation to the dietary stress. In other words, SEL flies pay a fitness cost late in life for the strategies used to survive the nutritional stress early in life, with signatures of selection at the genomic, transcriptomic, and metabolomic levels. Particularly for organisms like *Drosophila* that inhabit ephemeral and fluctuating environments, plasticity is thought to evolve to decouple the physiological needs that differ between adults and juveniles. However, here, through experimental evolution and 'omics rich data, Eroksar et al. provide new insights into the role of pleiotropy in shaping adaptation to early life stress.I really appreciated the multifaceted approach to understanding adaptation to nutritional stress early in life. The experiments are necessarily complex, and the manuscript is very rich in data-but, I had a difficult time throughout keeping track of how the different experiments linked to different results/conclusions/implications, etc.One issue is that plasticity wasn't well defined to begin in the introduction. The authors begin at line 46 with the statement "In contrast to such phenotypically plastic within-generation responses", but is not clear how the first paragraph showed phenotypic plasticity.

We have now modified the beginning of the 2nd paragraph (l. 52ff) to read:

" These physiological responses to developmental nutritional conditions are a form of phenotypic plasticity; i.e., a change of phenotype induced by differences in the environment, with no change in genome sequence (Scheiner 1993; Bateson, et al. 2004). We know much less on whether and how adult physiology and metabolism evolve genetically over generations in response to natural selection… "

We hope this clarifies the distinction between phenotypic plasticity and evolutionary change. What we state in the first sentence is a standard definition of phenotypic plasticity and the second reference cited there (doi:10.1038/nature02725) specifically discusses metabolic changes in adults induced by developmental nutrition as phenotypic plasticity.

The second paragraph suggests that one can examine plasticity by placing larvae in a different environment than adults.

No, this is not what we mean. In general, to study plasticity, one exposes individuals of the same strain or population to different environmental conditions and records the resulting differences in the phenotype. Here, we specifically talk about plasticity of adult phenotype in response to juvenile/developmental nutritional environment. Thus, to study it we need to let individuals develop under different juvenile nutrition treatments and study how this affect adult phenotypes; because we want to be able to attribute the effects specifically to juvenile conditions, the adults must experience the same environment. Of course, one could also study plasticity of juvenile phenotypes in response to juvenile nutrition, or of adult phenotypes in response to adult diets, but these are not a subject of this paper.

The second paragraph has been completely rewritten, so hopefully this should be clear.

However, this is also the selection regime, where only larvae are reared on nutrient-poor diets and adults are reared on the same standard diet.

Yes. The difference between phenotypic plasticity and evolutionary change in response to selection is not in the environmental factor (or experimental conditions) driving it, but in the underlying mechanism. We hope the beginning of the 2nd paragraph makes it clear now. We further explain how we operationally quantify plasticity in our study in the 1st paragraph of the Results.

Plasticity can be assessed in different ways (e.g., change in phenotypic variance, intragenotypic differences, reaction norms, etc.), and so a clearer definition would help ground the complexity of the following experiments.

We are looking at a change of trait means expressed in different environments, as we now state in the first paragraph of Results.

Further, does the starvation experiment count as plasticity in this framework? Clarification would help the reader understand and appreciate the complex experiments that are necessary to address this question.

Absolutely. We now added this sentence to the paragraph where we introduce the starvation treatment (l. 383): " The effect of the starvation treatment on the metabolic phenotype is also a phenotypically plastic response – to a different form of nutritional stress and one applied to adults rather than larvae. "

While the description of the five questions (lines 78-103) helps provide some context in how the reviewers are thinking about plasticity, it was disconnected with the subject material in the introduction and introduced further complications like pleiotropy. It might help by having a simpler statement in the introduction describing how plasticity can be assessed using the selected and control populations.

We have thoroughly rewritten the Introduction, with the new 2nd and 3rd paragraph explicitly introducing the general question of the plasticity-evolution relationship and the evolutionary independence or otherwise of larval and adult phenotypes. These provide the background for the specific questions formulated in the second half of the Introduction.

Another general comment is the error bar in Figure 3B, Figure 6B, etc. It is a little unusual to see just one error bar per panel in the figures. I also see that the error bar is the standard error of the mean-but mean of the main effect (evolutionary regime)?

The points in figure 3B (and the now removed figure 6) are marginal (= least square) means of metabolite abundances from the linear mixed model fitted to metabolite abundance data. The statistical model estimates the overall error variance and uses it as the basis for estimating the standard errors of each marginal mean. If the design is balanced (as is the case for these data), the standard error is the same for all means. We could put these SE bars on each point, but that would render the figure harder to read as there would multiple overlaps, possibly requiring adding some jitter to the points – and this would not add any information because the bars would be identical. We now added this text to Methods to explain this:

" To illustrate the effects of the experimental factors (in particular, the interaction between the effects of evolutionary regime and starvation; Figure 3B), for some metabolites we plotted the estimated marginal means from the GMM. Because the design was balanced and the error variance was estimated from the model, all marginal means had the same standard error; to reduce the clutter in the plots we plotted this common standard error as a single error bar rather than adding such bars to each symbol representing the mean."

I see the placement is zero centered for the log fold change, which is OK, but what is it for Figure 6B? An additional explanation would be appreciated.

Figure 6 has now been removed following a comment by reviewer 3. However, generally, metabolome results are typically zero-centered and scaled for the analysis; this is in part because the abundance estimates from the LC-MS approach we used cannot be meaningfully compared between metabolites. In contrast, the RNAseq data are normalized in a way that preserves the relative differences in the numbers of reads mapping to different genes, and thus allow one to distinguish genes with high and low expression.

Many of the experiments need additional rationale to describe them. While I got this to some extent in the introduction lines 78-103, the rationale and implications are underdeveloped for several of the key results.In the experiment associated with Figure 1, why are fat-body/carcasses used? It is described that the fat body is the key metabolic organ, but why wouldn't the whole body be used? For example, given the fecundity results presented later, how does the fat body transcriptome inform these results as opposed to what ovary or whole-body results would show? What is lost by focusing just on this one tissue group? If cuticular development and maturation genes are still most important, wouldn't that still show whole body RNAseq? Was the choice to enable linking the RNAseq with the metabolomics that follows? The rationale should be better clarified.

We aimed to focus on changes in the regulation of metabolism during the adult stage. Most of that metabolism takes place in the fat body, and this is in particular where dietary nutrients are converted into egg proteins and lipids, and where metabolic reserves of glycogen and fat are laid down or mobilized. While it is the ovaries that pack the vitellogenin and lipids into eggs, those key materials are imported there from the fat body. For these reasons we focused on the fat body for both gene expression and metabolome. By analogy, studies of the effect of nutrition on rodent metabolism often focus on liver (e.g. Agnoux et al. 2014, 2018; Safi-Stibler et al. 2020).

We now expanded the justification for focusing on the fat body to read:

" We focused on female abdominal fat body, the key metabolic organ combining the functions of mammalian liver and adipose tissue. It is in the fat body where metabolic reserves of glycogen and triglycerides are stored and mobilized, and where dietary nutrients are converted into the proteins and lipids subsequently transported to the ovaries for egg production (Li, et al. 2019). " (l. 156-160)

I'm also not sure I understood also why only the evolutionary regime differential expression results were shown, but not the larval diet results (summarized lines 162-188). I found the lack of parallelism an interesting result from the RNAseq experiment and would find them useful visualized (not as a table) at least in the supplement.

A similar point was made by Reviewer 1. We now added a plot illustrating GO-terms significant for the effect of larval diet (i.e., the plastic response) (Figure 1D), as well as a scatterplot of log-FC due to regime and larval diet (Figure 4A).

The comparison with Erkosar et al. 2017 was not well justified. In the methods, lines 614-15 it is mentioned that the larval gene expression is derived from an experiment that manipulated the microbiome, with and without microbes. The evolutionary regime was used as the main effect for comparison, but this isn't totally straightforward-why not just subset the 2017 data for only +microbe? why include the germfree at all here?

For the larval RNAseq in Erkosar et al. (2017), larvae derived from embryos that were first made germ-free by bleaching, and then mono-inoculated with 300 µl of suspension of a single strain of liquid culture-grown *Acetobacter* at OD=1 (microbe+) or treated with PBS (microbe-). This microbe+ treatment corresponds to inoculation with approximately 10^8 CFU per larval bottle. By contrast, in the present study we inoculated the embryos without bleaching and using 300 µl of suspension of feces at OD=0.5, which only contains about 10^3 CFU. Thus, in addition to a few other differences, the inoculation treatment used in the present study involved a much smaller microbial dose than the microbe+ treatment of Erkosar et al. (2017).

Furthermore, there is a trade-off in limiting the larval analysis to the microbe+ treatment because it cuts the number of samples by half, reducing statistical power. As a consequence, while the correlation across all genes between larvae and adults is the same whether we take all or only microbe+ data from larvae (r = 0.3522 vs 0.3518), the number genes that pass the FDR threshold is smaller; only 56 genes pass 10% FDR both for "+microbe" larvae and for adults. Still, 53 out of the 56 genes have the same sign of the effect (compared to 78 out of 84 genes when all larval samples are considered). Thus, the general result is robust to using only "microbe+" samples from Erkosar at al.

2017, but the signal is stronger when both "microbe+" and "microbe-" samples are used.

For those reasons we prefer to retain examining the similarities between the effects of regime on larval and adult expression by comparing main effects from the two experiments.

We added the following explanations to the Methods:

" All larvae were raised on the poor diet in that experiment, in either germ-free state or colonized with a single microbiota strain at the high concentration of about 10^8^ CFU (Erkosar, et al. 2017). Thus, the bacterial inoculation used for adult RNAseq (feces suspension containing about 10^3^ CFU) was intermediate between the two larval treatments. […] We used the main effect of evolutionary regime from this study for the comparison with the adult results; using only data from the microbiotacolonized treatment led to qualitatively similar conclusions."

Also, Erkosar 2017 data is from generations 177-200, which isn't stated here.

We added this information in the methods (it was around generation 190).

It's very surprising that the same genes are differentially expressed in mostly the same way between larvae and adult carcasses given all these differences in experimental design. The differences between these datasets should be explicit for the reader. However, given the complications listed above, I feel like the comparison between the carcass data and the Erkosar data does not add much to the overall message (and it's not visualized).

We would argue that the fact that a pattern is surprising also makes it interesting. The recommendation to remove this comparison goes in the opposite direction to what Reviewer 1 suggested; Reviewer 1 thought that the correlation between the effects of evolution on larval and adult gene expression was a one of the key results of our study, and recommend that we expand on it. This we did as summarized above, including adding an analogous correlation for the metabolome. As to the differences between the ways in which the larval and adult data were obtained (and the time elapsed between the two RNAseq studies): these potentially confounding factors would be expected to weaken the correlation. Thus, the fact that we see this positive correlation despite all the methodological differences implies that the true similarity of larval and adult changes is even greater. We make this point briefly in the Discussion (l. 476-480).

I liked the starvation experiment. Like before, it needs to be clear whether plasticity here refers to the std-poor diet and/or fed-starved comparison. It can be both but needed more information here. I think the authors consider the results to show more about the evolutionary regime (lines 373-375), but why isn't this also an aspect of plastic responses to nutritional stress? The interactions between fed-starved and the connection with differential allocation are very interesting results. While most of the discussion of this result is focused on the SEL resembling a starved metabolome, the change for SEL between fed and starved seems less than for the CTL. The CTL-poor seems to change most dramatically between fed and starved, which makes sense given the previous results of this selection experiment. Does this mean that SEL can buffer the nutritional stress better? I'm curious how buffering of this starvation stressor, despite previous evidence that SEL are less starvation resistant (line 331), informs our understanding of the adaptation to nutritionally poor larval diets.

Indeed, the response of the of the metabolome of Selected flies to starvation in the PCA is quantitatively smaller than that of Controls. However, to say that the metabolic state of Selected is more robust to starvation than that of Control, the starved Selected would have to look less like starved flies than starved Controls. This is not the case; rather, Selected and Controls converge to similar state upon starvation; it is in the fed state where they are different. We elaborated on this interpretation as follows (l. 427ff):

" Differential response of the Selected and Control populations to starvation is supported by regime × starvation interaction on PC1 scores (*F*_1,10_ = 10.7, *P* = 0.0084) – the metabolic state changed less upon starvation than that of Controls. This does not imply that the metabolome of Selected flies is more robust to starvation. Rather, fed flies from the Selected populations were situated closer to starved flies along the starvation-loaded PC1 axis than fed Control flies (Figure 4A; *F*_1,17.8_ = 14.0, *P* = 0.0015, GMM on PC1 scores); in the starved condition they converged to a similar state (*F*_1,17.8_ = 0.0, *P* = 0.99; see Supplementary file 5 for detailed statistics)."

Great to see some overlap between genomic, transcriptomic, and metabolomics in purine synthesis.

Thank you, this is why we think that figure 3 is useful in putting these overlapping results on a common map.

Please make clear that the Kawecki 2021 genomic data is from ~150 generations.

Done (l. 255).

For the adult costs of larval adaptation, it is interesting that CTL laid more eggs than SEL populations when exposed to a poor larval diet. Would you see the same result if fecundity was assayed for both on the standard diet? i.e., does the evolutionary regime just lead to generally fewer eggs produced by SEL populations?

We have already published that, after 64 generations of experimental evolution, the

Selected populations had lower fecundity than Controls when both raised on standard larval diet (Kolss et al. 2009). This, however, can be interpreted as a consequence of the Selected specializing on the poor larval diet at the expense of performance when raised on standard larval diet. We have seen other signs of this, such as somewhat reduced larval viability and substantially smaller adult size of Selected compared to Controls when raised on the standard diet. This kind of trade-off, whereby populations evolving under new conditions lose somewhat in terms of performance in the ancestral conditions is rather expected. So, what we find here indeed suggests that Selected have a generally lower fecundity, suggestive of an larval-adult trade-off (rather than poor larval diet – standard larval diet trade-off). We added a shorter version of this point to the Discussion (l. 496-498):

"We have previously reported a lower fecundity of Selected females compared to Controls when both were raised on standard diet (Kolss, et al. 2009); we interpreted that finding as a manifestation of reduced performance of Selected populations when raised on the standard larval diet."

Does it matter for adaptation to nutritionally poor diets if SEL populations are just less fecund? I think the rationale needs some additional justification here.

Absolutely; eggs to breed the next generation are sampled haphazardly from those laid by the females in a setup identical to that used to quantify fecundity. Thus, a female that lays more eggs will contribute more offspring to the next generation. We now state this explicitly (l. 500-504):

" Other things being equal, a female's contribution to the next generation under the experimental evolution regime is proportional to the number of eggs she lays within a time window similar to that used to quantify fecundity. Thus, the fecundity reduction we found in Selected adults represents a significant fitness trade-off of the improved ability of the Selected larvae to survive and develop under extreme nutrient shortage."

Additionally, in Figure 5, can these be visualized in a way that shows the true range of observed values (e.g., boxplots)?

This is now figure 7. We were not comfortable with using a boxplot because it would conflate two levels of replication (replicate populations and several replicates per population). Furthermore, the nature of the points differs among the three plots: for ovariole each data point is one female, for eggs number if it is the number of eggs laid by a group of females divided by their number, and for egg-to-ovariole ratio it is the average number of eggs divided by the average number of ovarioles for each population. So only the ovariole data represent values for individual females. Therefore, we retained the bar graph but, to illustrate the variation among populations, we added points that represent the means of each population.

I think the discussion provides important context, and the authors highlight one potential explanation for the results is pleiotropy across life stages. This is an interesting perspective given that *Drosophila* are holometabolous insects, and as the authors say, this remains surprisingly understudied. It's not clear whether this cross-life-stage pleiotropy is amplified because of holometabolous insects or also generalizable to other organisms as well.

While this is not clear empirically, an argument has been made that a key advantage of insect complete metamorphosis is that it eliminates such pleiotropy. Thus, in organisms without metamorphosis such pleiotropy would be more likely. To make this point, we now added the following text to the Discussion (l. 582ff):

" A major advantage of this complex development is thought to be that it decouples larval and adult gene expression, promoting independent evolution of larval and adult phenotypes (Moran 1994; Rolff, et al. 2019). Contrary to this notion, our results suggest that, at least on the scale of hundreds of generations, evolutionary changes in physiology driven by selection acting on larvae may have maladaptive pleiotropic effects on adult physiology. If this is the case in a holometabolous insect, such evolutionary non-independence of juvenile and adult physiology would likely be more pronounced in species with more developmental continuity between juvenile and adult tissues and organs. "

Reviewer #3 (Recommendations for the authors):Erkosar et al. use gene expression and metabolomic analyses to investigate the evolutionary change that accompanied over 230 generations of selection by nutrient deprivation from egg to pupation in *Drosophila*.The evolved populations were six replicate populations, each selected for larval development and growth under poor nutrient conditions. Each population was kept at ~200 flies at each generation, which may have allowed significant drift when compared to larger populations in similar experimental evolutionary studies however, this regime has resulted in repeated evolution of a few interesting traits. These traits were revealed in earlier work and include faster developmental time, higher egg-to-adult viability, and smaller adult size. Additionally, selected lines showed higher sensitivity to starvation as adults and the allele frequency changes that accompanied selection intersect with those loci associated with selection for starvation resistance.The picture painted from previous work is of genotypes arising from selection that are more likely to make pupae/adults; to make them faster and more numerously than the control population. The adults that they make however are not as healthy as the adults from the control population. This certainly suggests that adult traits suffer a constraint based on the evolution of juvenile condition. Is that constraint manifest in patterns of gene expression, or in patterns of variation in the metabolome? Hints at metabolic adaptation within these populations came from a prior publication that described a difference in nitrogen or carbon assimilation between these populations, which is at least intriguing.The current work is important in that it sheds light on the kinds of endophenotypic changes that accompany these evolutionary changes. In addition to the condition-specific patterns of expression and metabolite abundance, the authors also test their ideas about how the transcriptome and metabolome of adults of both populations respond to diet, and if there is a resemblance to flies evolved on poor food and starved flies.This paper presents several interesting results. One, the evolution that occurred in the larval transcriptome was largely similar to the differences between the populations as adults. This implies a constraint in adult phenotypes on juvenile phenotypes and vice versa that is reflected in the transcriptome.Within the same experiment, the transcriptome and metabolome of adults from both populations appear to respond similarly to diet. That is, the adults of both populations, despite significant shifts in metabolome and transcriptome, show very similar responses to being placed on the poor diet, when compared to the normal diet. This is an unexpected result given the constraint that may explain the previous result.That the within-generation response to diet appears independent of the evolutionary response to diet conditions is interesting. The authors could develop their discussion of these results further. I think it is worth considering that the effects of diet with a generation are not necessarily adaptive. An adult on a poor diet might have a metabolome that looks a certain way, reflecting in part the exhaustion of resources. The adult from a larva that has evolved on a poor diet however may more closely approximate an adapted metabolome, there is really no reason to think that these would be the same.It is also interesting that the effect of diet on the adult metabolome was somewhat similar to the difference between the adult metabolome of the two populations, the evolutionary change. This is in contrast to the seemingly independent effect of diet on the transcriptome, and the transcripts that differ between the populations. I hope that this result is not a 'false positive' due to the small biological sample size of the transcriptome data (n=1) relative to the metabolome data (n=2), or a 'false negative' due to the relatively sparse metabolome coverage (~100 metabolites) that is typical of targeted analysis of fly metabolome compared to the more comprehensive transcriptome. This result is in line with other studies that indicate that endophenotype appears to converge as it goes from: gene > RNA > protein > metabolite, to phenotype. The relative anticipation of the evolutionary response seen in the metabolome differences by diet when compared to the same in a relationship in the transcriptome is neat.The results described above manifest largely in PCA, in reduced dimensional space, which is appropriate and telling, but may hide variation that occurs within and between populations, and across diets. There is some agreement within the univariate analysis of metabolites and transcripts that are differentially abundant between populations and treatments, suggesting that the 'biological signal' seen in the PCA could be picked apart using the identities of the genes and metabolites. The authors indulge in this pursuit and while creative, take their interpretation too far. The authors make several clever interpretations of their data in order to shed light on potential mechanisms (pathway activity) that may explain their results. Without testing some of these ideas however this amounts to speculation.

We have reduced speculation about potential meaning of changes in metabolite abundance and gene expression at specific pathways, but we still think we are justified to identify pathways (in particular amino acid and purine metabolism) given multiple signals from RNAseq and metabolome at both larval and adult stage.

The third point of interest is the "Metabolic profile of Selected flies tends towards starved-like state", an argument that the metabolome of selected flies more closely resembles starved flies than does the metabolome of control flies. This is argument rests on MANOVA and some agreement in sign with metabolite levels upon starvation, and those affected by selection. This is an interesting result that could relate to the sensitization to adult starvation in the evolved population. At the univariate level however, the picture is quite complicated (Figure 3B). The interactions between diet, selection regime and the effect of starvation are complex. The authors make a valiant effort at interpretation, however, I don't think their conclusions are well supported from these data. I suggest approaching the univariate analysis more circumspectly.The authors are over-interpreting the meaning of their data. This is my main objection to what is otherwise an interesting and meaningful study. I think that the authors should include their interpretation of the data, however, it should be presented as speculation, rather than as if the interpretation were itself an observation.

We modified the text to state clearly that not all metabolite groups show a similarity between the effects of evolution and starvation. We also modified the language to emphasize the speculative nature of these interpretations ("appear consistent with…", "we have not demonstrated a direct link…", "it is tempting to speculate…"). We think it should be clear to the reader that much of interpretations of our data are indeed speculative in the absence of independent experimental tests.

The joint analysis presented in 'Figure 4—figure supplement 1' is problematic for several reasons. I don't think it should be included as it does not add anything to the paper worth discussing.

We have now removed this analysis.

"[abstract line # 26]: "resulting in deficiency of electron transports and congestion in β-oxidation." Two objections, first, 'deficiency' is different from 'reduced', we have no idea if the reduced level of these RNAs is at all affecting the trait, these are associations and cause and effect is not known. Second, RNA is not protein, and even then, protein abundance is not activity, so either discuss these results in a more speculative light or make a more direct and independent test of electron transport activity/function. Further, congestion in β oxidation is certainly a possible explanation for the patterns of acetylcarnitines and other FA-derivatives that are more/less abundant, however, this too is a speculative explanation and ignores the equally valid evidence that this explanation is wrong.

We have now removed any speculation about the significance of differences in NAD, FAD and acylcarnitines.

Similarly, the section that discusses the amino acids [#492-505] describes some amino acids as 'deficient' in the selected flies. These amino acids are reduced in abundance compared to the control flies, but this is not the same as deficient. Deficiency implies that if there were more of these then the outcome would change. That result is unknown. The data are describing an association with abundance, concluding that this is a deficiency is misleading.

We meant to use "deficiency" in a relative sense, analogous to the way "underexpression" is used for genes showing a lower expression in the focal condition than in some other condition. We changed this to "underabundance" or "lower abundance" throughout the ms.

In line #385, "This analysis confirmed that the changes in amino acid and purine metabolism contributed to adaptation to poor larval diet." Confirming that something contributes, implies that you have gone beyond association and tested causality. These data do not demonstrate cause or effect but bolster a previously-identified association.There are two sections of the results (lines 217-290, and 378-404) dedicated to the pathways that might be involved in the transcriptome and metabolome differences seen in these populations. I suggest paring this down, at least as a part of the Results section, as enough of it is speculation. Lines #378-404 are mostly speculative and so this argument should move to the discussion.

We took care to purge these parts of Results of any speculation as to potential causes or implications of the differences between Selected and Control populations in metabolite abundance. Still, while it could be argued that Figures 1 and 2 could stand on their own, we thought that it would be useful to point the reader to specific patterns in the data, and to briefly explain the role of those compounds in metabolism to less initiated readers. The former lines 378–404 (now l. 250-272) report the joint pathway analysis of Metaboanalyst, as well as describing an overlap between the data for gene expression, metabolome and genomic differentiation for the key pathways. It would be awkward to place it in Discussion. We also note that Reviewer 2 made a specific appreciative comment about this part ("*Great to see some overlap between genomic, transcriptomic, and metabolomics in purine synthesis*.").

Similarly, the discussion could be pared down given some of the contradictions that emerge when the data are discussed. For instance, if the paragraphs from lines #492-505 resolves with the axiom: "the differences in amino acid abundance between Selected and Control populations is consistent with differential use of amino acids in metabolism – their interconversion, catabolism and use in the synthesis of other metabolites.", then I'm not sure that it's a good use of space.

We are unsure which contradictions the reviewer alludes to. The statement quoted by the reviewer is not an "axiom"; it is one of three potential general reasons for differences in abundance of free amino acids. The other two are (i) a change in their supply from nutrition and (ii) a change in the rate of their use for protein synthesis. The point we are making in this paragraph is that the pattern of differences in amino acid abundance between Selected and Control flies is not consistent with the two latter explanations, leaving the first explanation as the more likely one. We rewrote this paragraph in an attempt to make this point clearer and also removing the somewhat tedious discussion of the abundance of specific amino acids in the diet relative to protein synthesis need (l. 517-524).

Figure 6 could be idealized (made with generic genes) or excluded. I say this because there are at least 10,000 genes from which to pull patterns like these from, so seeing these genes like this does not have meaning (FDR is high). The idea behind this figure can certainly be discussed as it is in the discussion, but this data set is underpowered for this kind of interference.

We removed this figure.

The authors use the term pleiotropic at several points. I think they are referring to something like antagonistic pleiotropy. I think a short section that introduces this term in this way, with reference to the theoretical or experimental work on the idea in relation to the evolution of aging perhaps would help the reader.

This is an excellent point and we now elaborate on the analogy to the early-life inertia theory of aging as well as to "intralocus sexual conflict" over gene expression at the end of Discussion (l. 590ff).

In the bigger picture, there are 'inward' and 'outward' implications for this work. The authors present support for their conclusion that negative trade-offs in adulthood are tied to the endophenotypic changes that occur in juveniles as a result of selection when only applied to juveniles. This support comes from the within-population similarity in adult transcriptome and metabolome to that of larvae. The 'inward' perspective, while tantalizing, is perhaps over-presented in the discussion of these results. Insight into the mechanisms (cellular pathways) by which these trade-offs happen can and should be included, however there are also wider implications of these results for insect evolution, the 'outward' perspective. These results suggest that the evolution of juvenile phenotypes is constrained by the outcomes they associate within the adult, at least on juvenile nutrition inputs/needs. The authors cite some theoretical papers on the topic, and I think the reader would be better served if the authors could reflect on the implications of the constraint they appear to have found. What does this imply for the phenotypic (and environmental) space that the juveniles are constrained to? We know that *Drosophila* are very choosy when it comes to oviposition, and here we see one of the reasons why that might be the case. Do the authors think that adult choice has evolved in part to deal with this constraint? It's easy to see the negative effects, the burden, of such a constraint. Evolution is powerful though, so why then is such constraint 'allowed'? Could this constraint actually reflect some kind of as-yet unconsidered benefit? For instance, does a constrained gene-trait map allow Drosophila to maintain a smaller, simpler, genome? I am not familiar with the literature on this and other topics that come to mind when the weight of these results are realized. Perhaps the authors could spend some of the discussion to share their interpretation?

We have restructured the discussion in part in response to these comments. We finish it with a section that speculates on potential implications of the constraint, as well as placing it in the broader context of other evidence of constrains on regulatory evolution, such as those thought to be implicated in aging and sexually antagonistic pleiotropy. We did not venture into discussing *Drosophila* oviposition behavior or genome size; we feel anything we could say about this would be far too speculative and tangential to the topic of the paper.

There are a few passages that I found hard to understand. When describing part of their hypotheses about the evolution of adult and juvenile phenotypes, they write [#82-85]: "Plastic responses that program metabolism adaptively for a poor nutritional environment in adulthood would be misdirected in the Selected populations because the adults were switched to standard diet; evolution should thus counteract them." Why would plastic responses in adults be affected by selection in juveniles? If they were affected, why would we call them plastic?Keep in mind that this sentence is followed by [#85-87]: "Conversely, plastic responses that alleviate the consequences of having developed on a poor diet irrespective of adult diet should have become amplified as a result of evolution on the poor diet." It sounds like the authors are predicting that two counteracting effects have both happened and this does not make sense to me. I would like to understand the authors' point here and I would hope that it could be clarified in a revision.

This text has now been removed. We now introduce the subject of plasticity versus evolution in more general terms in of the second paragraph of the introduction.

There were several references to the figures that don't seem correct, and other parts of the figure referencing scheme that did not make sense. For example, the reference to 'Figure 2' on line #231, refers to a PC plot, and not the heatmap shown in Figure 2. Line #231 refers to Supplementary Figure file 4, and I'm not sure that's correct. Please make sure that the figure references point the reader to the correct figure and that the figures are referred to and arranged in numerical order.

We apologize for these mistakes in the previous version. We have now double-checked all references to figures and supplementary files.

The experimental design, the sampling scheme and batch design of the analyses were all good. These experiments made the most of the number of samples that were used. The statistical analysis as described is appropriate, however there is enough going on that making the code available would probably help raise the impact of this work.

We appreciate the reviewer's trust in our analyses. We understand the reviewer suggests to make the code available because it might be useful for other researchers. However, making our diverse scripts sufficiently user-friendly and annotated to be useful to others would require a very large amount of work. And even if we did this, the usefulness to other people would be limited. Nothing that we do in terms of statistical analysis is non-standard or novel. In part, we do basic statistical analyses such as PCA, Pearson's correlation, MANOVA or general mixed models, which are implemented in almost any statistical package. We used mostly SAS statistical system, which is now used by a relatively small number of researchers in biology, further reducing the general interest in the scripts. In the other part, we follow standard pipelines in published packages designed for special analyses (RNAseq read filtering and mapping, differential expression analysis, GO-term enrichment analysis). Thus, we are not convinced that including all our scripts would be useful to the readers or would increase the impact of the paper.

The authors are clearly aware of the multiple testing problem and I stand by my recommendation to remove figures that make use of data that do not survive FDR correction.

The philosophy behind the FDR correction used in the omics approaches is that the statistical test on each gene or metabolite tests a different biological hypothesis. We agree that this approach is essential when screening for candidate genes or metabolites, and we do not make any statement about any specific gene or metabolite being differentially expressed/abundant if it does not pass the FDR threshold. However, the number of genes passing an FDR threshold typically greatly underestimates the number of genes that are actually affected (i.e., there are more false negatives than false positives). This is also the case for our data, based on the estimated numbers of true nulls. Therefore, when examining broad patterns of expression changes, such as we do when comparing adult and larval expression or metabolite abundance changes, we went beyond the rather few genes/metabolites that passed the FDR threshold at both stages, and also looked at other genes with nominal P < 0.05, or even at all genes. In this approach, each gene / metabolite is treated as a point estimate and we are not making conclusions about any specific gene / metabolite but about the overall pattern.

A point I will need clarification on occurs in lines [#601-604]: 'Therefore, we also estimated the number of genes that differed in expression due to each factor in the analysis as the total number of genes minus the estimated number of "true nulls" (Storey and Tibshirani 2003) (https://bioconductor.org/packages/topGO/), using the weight method and Fisher's exact test.' What does this mean? By 'factor in the analysis', I assume you mean regime and diet. Why, and how, would you use topGO to evaluate the number of genes affected by regime or diet?

It seems somehow a piece of text got translocated from the end of the preceding paragraph on GO-term enrichment analysis; we apologize for this and thank you for spotting it. We corrected this so the offending sentence (now l. 689-691) now ends after the citation as intended: